# Single-gene resolution of diversity-driven overyielding in plant genotype mixtures

Samuel E. Wuest [1,2,3,4] ✉, Lukas Schulz[5], Surbhi Rana[6,8], Julia Frommelt[1], Merten Ehmig[7], Nuno D. Pires[2], Ueli Grossniklaus [2], Christian S. Hardtke [6], Ulrich Z. Hammes[5], Bernhard Schmid [1,3] & Pascal A. Niklaus [1]

In plant communities, diversity often increases productivity and functioning, but the specific underlying drivers are difficult to identify. Most ecological theories attribute positive diversity effects to complementary niches occupied by different species or genotypes. However, the specific nature of niche complementarity often remains unclear, including how it is expressed in terms of trait differences between plants. Here, we use a gene-centred approach to study positive diversity effects in mixtures of natural *Arabidopsis thaliana* genotypes. Using two orthogonal genetic mapping approaches, we find that between-plant allelic differences at the *AtSUC8* locus are strongly associated with mixture overyielding. *AtSUC8* encodes a proton-sucrose symporter and is expressed in root tissues. Genetic variation in *AtSUC8* affects the biochemical activities of protein variants and natural variation at this locus is associated with different sensitivities of root growth to changes in substrate pH. We thus speculate that - in the particular case studied here - evolutionary divergence along an edaphic gradient resulted in the niche complementarity between genotypes that now drives overyielding in mixtures. Identifying genes important for ecosystem functioning may ultimately allow linking ecological processes to evolutionary drivers, help identify traits underlying positive diversity effects, and facilitate the development of high-performance crop variety mixtures.

Numerous experiments in natural and managed ecosystems including grassland, forest, and cropland have shown that productivity often increases with the diversity found in plant communities[1–6]. Diversity is typically quantified as the number and abundance of species (species diversity) or the occurrence and variation of specific plant trait values (functional trait diversity)[1,6–12]. It is assumed that niche complementarity underlies biodiversity–ecosystem functioning (BEF)

relationships, but this is difficult to demonstrate empirically because niches are difficult to quantify[6,8,13–17]. In functional ecology, trait differences are often used as an indicator of niche differences[18–22], and are major determinants of the composition, diversity, and functioning of communities[18,23–25]. However, it is currently less clear which trait differences drive the positive BEF relationships in plant communities[17,25–27], but see [22,28–31]. The trait-based approach also has

[1]Department of Evolutionary Biology and Environmental Studies and Zurich-Basel Plant Science Center, University of Zurich, Winterthurerstrasse 190, 8057 Zurich, Switzerland. [2]Department of Plant and Microbial Biology and Zurich-Basel Plant Science Center, University of Zurich, Zollikerstrasse 107, 8008 Zurich, Switzerland. [3]Department of Geography, Remote Sensing Laboratories, University of Zurich, 8057 Zurich, Switzerland. [4]Agroscope, Group Breeding Research, Mueller-Thurgau-Strasse 29, 8820 Waedenswil, Switzerland. [5]Plant Systems Biology, School of Life Sciences, Technical University of Munich, 85354 Freising, Germany. [6]Department of Plant Molecular Biology, University of Lausanne, Biophore Building, Lausanne 1015, Switzerland. [7]Department of Systematic and Evolutionary Botany, University of Zurich, Zollikerstrasse 107, 8008 Zürich, Switzerland. [8]Present address: Department of Crop Genetics, John Innes Centre, Norwich Research Park, Colney Ln, Norwich NR4 7UH, United Kingdom. ✉e-mail: samuel.wuest@agroscope.admin.ch

some limitations, in particular when it comes to understanding positive biodiversity effects: first, traits often co-vary because of evolutionary trade-offs between ecological strategies[32,33], which makes it difficult to distinguish correlation from causation in trait-based analyses of BEF experiments. Second, while traits often are associated with environmental conditions[19,34] (e.g. correlation between specific leaf area and soil moisture), it remains unclear whether differences in these traits drive the studied diversity effect, or whether other, unknown trait differences that correlate with these underlie the observed effects. Finally, it also is possible that many small phenotypic trait differences need to be considered simultaneously to adequately capture niche complementarity between plants[26,35], making it more difficult to identify specific mechanisms that cause BEF relationships[1,36]. Here, we explore a gene-based approach to investigate the causes of positive diversity–productivity relationships in plant stands. Our new approach complements traditional trait-based methods and helps to identify causal drivers.

Positive BEF relationships occur not only at the interspecific but also at the intraspecific level; for example, mixtures of genotypes of wild plants and crops often overyield relative to monocultures of the same genotypes (see, e.g.[9,11,12,37–39]), although there are exceptions[40]. The general mechanisms underlying niche complementarity and overyielding may be similar in both cases, although the potential for niche differences between species is greater than between genotypes of the same species. Here, we focus on positive genetic diversity effects in plant stands of the model plant *Arabidopsis thaliana* and compare

genotype mixtures that contain either one or two alleles across regions of the genome. A major advantage of this approach is that the diversity of traits and alleles can not only be manipulated by assembling different plant stands from an existing pool of genotypes, but also from new genotypes created through crosses (Fig. 1). Crosses allow, within the limits of linkage disequilibrium, a redistribution of genetic variation, and consequently trait variation, between genotypes. The assembly of new plant stands that differ in their genetic composition then allows establishing causal links between genetic diversity and the properties of mixtures[41,42] (Fig. 1). Several recent papers have expanded the traditional approach that links genetic differences amongst individuals to their phenotypic variation to the genetic study of the properties of ecological communities[41–45]. For example, and in analogy to keystone species that exhibit disproportionately large effects on ecosystems, Barbour and colleagues describe a plant "keystone gene" whose presence determined the stability of an experimental trophic food web containing plants, aphids, and their parasitoids[46]. Together, these publications demonstrate that genetic effects can cascade across layers of increasing biological complexity, sometimes in unexpected ways.

In the present work, we conduct a genetic study on how allelic diversity affects the overyielding of genotype mixtures and combine this with experiments investigating how the identified allelic variation affect biochemical and physiological functions of the plants. We find that mixture overyielding is driven by allelic diversity at a single, major-effect quantitative trait locus (QTL), and use association mapping to

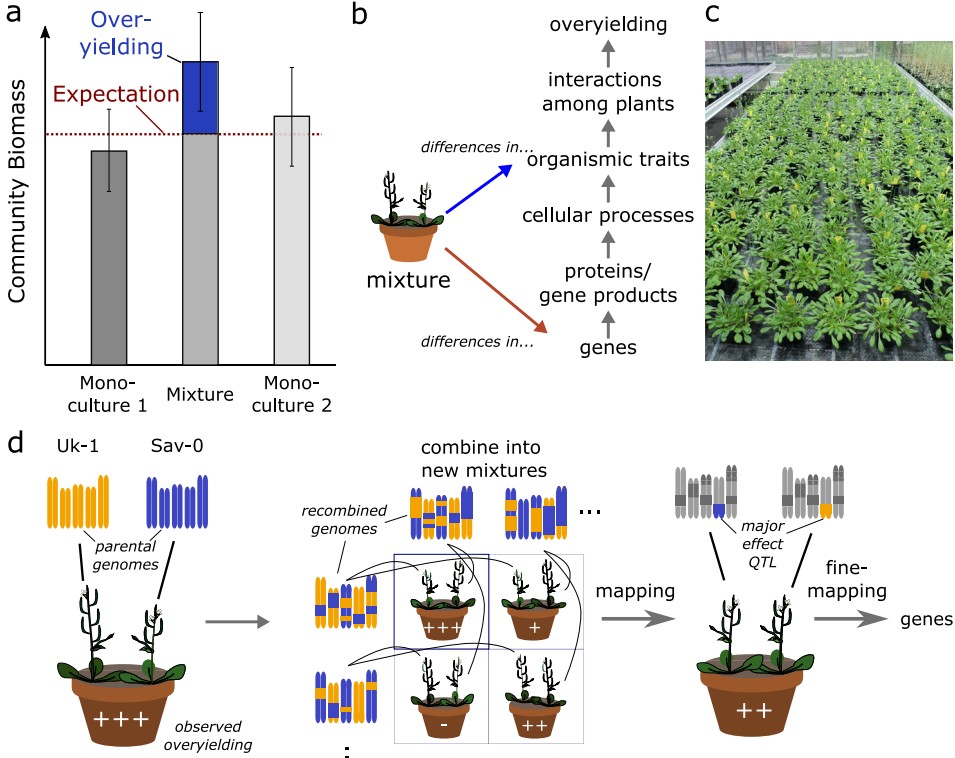

**Fig. 1 | Experimental approach to the genetic dissection of positive diversity effects. a** A positive diversity effect (blue) in pair-wise mixtures denotes the estimated deviation of mixture yield from expectations based on monoculture yields (overyielding). **b** Different possible paths for establishing the drivers of diversity-driven overyielding in mixtures. Past work has put much effort into identifying the underlying phenotypic trait differences (blue arrow), but our work is concerned with studying the underlying genetic differences (red arrow). It may then become possible to infer relevant functional trait differences from genes, i.e., to move up through the intermittent levels of biological organisation. **c** Experimental setup used in this study, showing model plant stands consisting of

four plants and different pairwise genotype combinations. **d** Schematic representation of how a genotypic diversity effects (left; Uk-1 + Sav-0, exhibiting many genetic differences across the five chromosomes) can be further dissected into specific genetic diversity effects through genetic mapping, i.e., with the use of crosses and genetic recombination followed by the assembly of genotype pairs into new mixture compositions. "+" (or "−") denote mixture performance higher (or lower) than expected. In the scheme, a major-effect QTL explains a large proportion of diversity-driven overyielding in the parental mixture, so that overyielding occurs in mixtures of near-isogenic lines that only contains this "functional" genetic diversity.

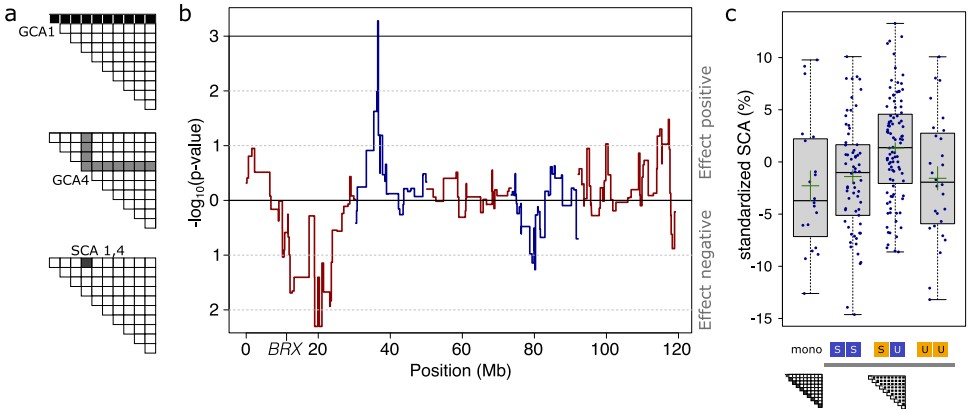

**Fig. 2 | Genotypic and allelic diversity effects. a** Illustration of the concept of General and Specific Combining Ability (GCA and SCA) derived from different genotype compositions assembled according to a competition half-diallel design. GCAs of genotypes 1 and 4 are estimated from productivities of all mixtures in which these genotypes occur, $SCA_{1,4}$ denotes the estimated productivity deviation of communities containing these two genotypes after accounting for GCAs. **b** QTL map of allelic diversity associated with variation in SCA within genotypic mixtures. Blue and red lines denote the different chromosomes. "*BRX*" indicates the location of the *BREVIS RADIX* gene. **c** Boxplots showing SCA distributions of different plant stands: genotypic monocultures (mono; $n = 20$), genotypic mixtures but allelic monocultures at the QTL on chromosome 2 (SS; $n = 66$ and UU; $n = 28$), genotypic mixtures and allelic mixtures at the QTL (SU; $n = 96$). Green lines denote mean values +/− s.e.m. Boxes show interquartile ranges with medians; whiskers indicate data ranges up to 1.5 times the interquartile range from the box. Source data are provided as a Source Data file.

resolve it to variation at the *AtSUC8* gene. Genetic variation at this locus is also associated with altered root growth responses to changes in substrate pH, and results in altered protein function. This shows that mixture overyielding can have a surprisingly simple genetic basis, and that the genetic approach can yield new insights into the causes of positive diversity effects.

## Results

### Umkirch-Slavice mixtures consistently overyield

In order to genetically dissect the mechanisms that underlie biodiversity effects on productivity, we first identified genotypes that overyielded when grown together in mixture, i.e., genotype combinations that produce more biomass than the average of their monocultures (Fig. 1a). We tested overyielding in ten mixtures, each containing one pair of *Arabidopsis thaliana* (L.) Heynh. genotypes. (Supplementary Fig. 1a). We used these pairs because they are the parents of publicly available recombinant inbred lines, a convenient resource for genetic studies. All overyielding estimates from this experiment were not statistically significantly different from zero. This was not unexpected, because overyielding is calculated as difference between three yield values (of the mixture, and of the two monocultures); a high replication of all three communities is therefore required to compensate for error propagation in this calculation. However, mixtures that contained the two accessions Slavice-0 (Sav-0) and Umkirch-1 (Uk-1) had consistently higher average yields than the monocultures across all substrates and pot sizes. We verified this overyielding in a second experiment with two different pot sizes and two planting densities (Supplementary Fig. 1b). Across all experimental settings, mixtures of Sav-0 and Uk-1 yielded an average 5.6% (range: 0–12%) more biomass than expected based on monoculture productivities, and a meta-analysis combining all experiments revealed that this estimate was significantly different from zero ($P = 0.012$). This effect is relatively large for a pot-based within-species diversity experiment. For comparison, the average overyielding in field trials with crop variety mixtures typically ranges from 2 to 4%[12,38,47,48].

### A single major-effect QTL promotes overyielding through an increased complementarity effect

To study the genetic basis of this effect, we established a competition diallel panel (Fig. 2a), an experimental design in which genotypes are systematically combined in all possible pairwise combinations[49–52]. In

diallel analyses, general and specific combining abilities (GCAs and SCAs, Fig. 2a) can be taken as proxies for additive and non-additive mixing properties of genotypes and genotype combinations. GCAs capture the average additive contribution of a genotype to the productivity of mixtures in which it occurs. SCAs capture the productivity deviation of a specific genotype mixture from expectations based on the additive contributions of the components (the sum of the genotypes' GCAs). Here, we used a half-diallel design containing 18 randomly selected recombinant inbred lines (RIL) derived from a cross between Sav-0 and Uk-1, and the two parental lines. These RILs had been created to allow the map-based cloning of the *BREVIS RADIX* (*BRX*) gene, in which natural variation causes strong differences in root architecture between Sav-0 and Uk-1[53]. We hypothesized that such differences could drive overyielding in genotype mixtures, which, however, turned out to be wrong. The 20 chosen genotypes were grown in all pair-wise combinations and genotype monocultures were grown in duplicates. The diallel was replicated four times at different dates (temporal blocks), resulting in a total of 920 pots sown for this experiment. We further used two different substrates (sandy and peaty soils, two blocks each, with 230 sown pots per block). For each genotype composition, we determined aboveground dry matter production. Comparing mixture productivities to monocultures of the mixture components, we estimated that the 190 genotype mixtures overyielded on average by 2.8% (two-sided *t*-test, $t_{189} = 4.74$, $P < 0.001$). This estimate is slightly below the overyielding observed in the parental mixture. However, if the genetic differences driving overyielding are restricted to specific genetic loci, then only part of these genotype mixtures will overyield (the bi-allelic genotype mixtures), while the others will not. Therefore, and in order to genetically map such diversity-effect loci, we determined the average SCA across the four blocks for each of the 210 genotype compositions (190 genotype mixtures plus 20 monocultures). To adjust for differences in community productivity between substrates, and to obtain a normal distribution of residuals, we scaled the estimated SCAs by dividing these by the average community biomass on the respective substrate. SCA thus was expressed as effect relative to the mean productivity of all compositions on the substrate. Next, we tested whether variation in SCA among genotype compositions could be attributed to genetic differences at specific genomic regions. Since the published marker density for the RIL population used here was relatively low, we first constructed high-resolution genotype maps by whole-genome re-

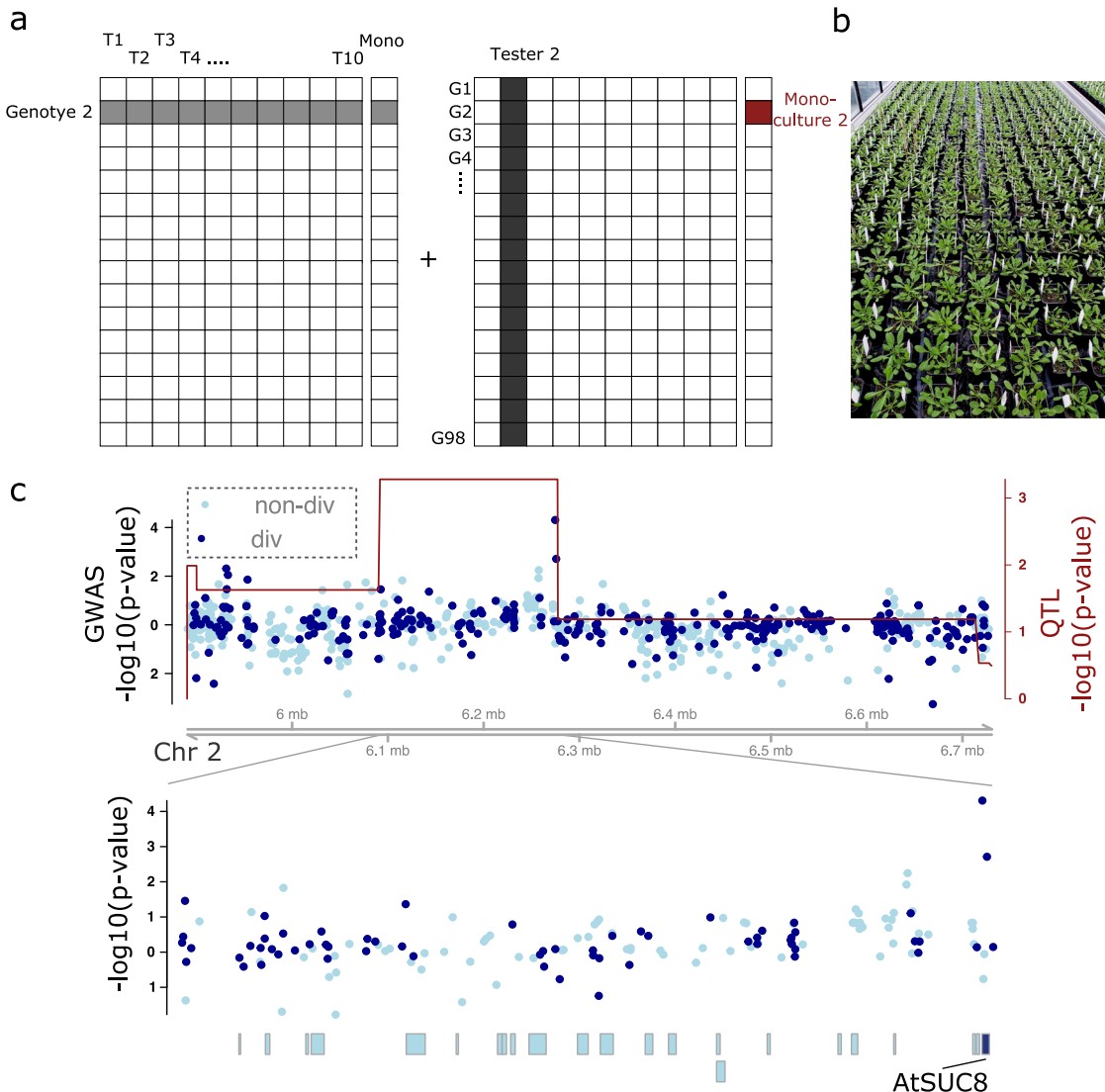

**Fig. 3 | Single nucleotide polymorphism differences at the *AtSUC8* locus associate with positive diversity effects in genotype mixtures. a** The experimental design represents a fully factorial combination of ten tester genotypes with each genotype of a panel of 98 natural *Arabidopsis* accessions. **b** Picture of the experiment. **c** The QTL mapping results (red line and right axis) overlaid with the genetic association results (blue dots and left axis). Light blue dots denote SNPs at which the Sav-0 and the Uk-1 tester lines do not differ (non-div), dark blue dots denote those at which they do differ (div). Dots above zero indicate positive diversity−SCA associations, dots below zero negative ones. Boxes in the bottom panel denote gene regions, the *AtSUC8* gene region is colored dark blue. Source data are publicly available through the Zenodo data respository (DOI: 10.5281/ zenodo.6983283).

sequencing of each line (Methods, Supplementary Fig. S2a). We then used marker-regression to compare SCAs of mixtures that were either mono-allelic or bi-allelic at a given marker region, i.e. we tested for effects of allelic diversity. We found that SCAs were positively associated with genetic differences at a single quantitative trait locus (QTL) on chromosome 2 (Fig. 2b). The high-density marker map allowed us to resolve this QTL to a very small genomic region, spanning approximately 178 kb. Mixtures that exhibited allelic diversity in this region had a 2.8% (+/− 0.8% s.e.m.) higher SCA than mixtures that contained only one of the two alleles ("mono-allelic" genotype mixtures, Fig. 2c; contrast between mono-allelic and bi-allelic genotype mixtures $t_{187} = 3.53$; $P < 0.001$). The SCA of the mono-allelic genotype mixtures (i.e., mixtures of genotypes which contained only the Sav-0 or only the Uk-1 allele at the identified QTL on chromosome 2) averaged 0.8% higher than the SCA of genotype monocultures, but this difference was not statistically significant. We then applied the additive partitioning method[54] to test whether overyielding within genotype mixtures (i.e. contrasting bi-allelic with mono-allelic mixtures) was associated with

the dominance of productive genotypes carrying either the Sav-0 or Uk-1 allele (a so-called "selection effect", SE), or whether both allele carriers benefitted from growing in mixture (so-called "complementarity effect", CE). We found that overyielding was largely driven by CEs (contrast between mono-allelic and bi-allelic mixtures $t_{187} = 2.57$, P = 0.01) and not by SEs ($t_{187} = 0.201$, $P = 0.84$). The average CE was 24.7 mg, which corresponds to 3.1% of the biomass average of all genotype compositions analyzed in the experiment. Overall, the experiment shows that genotypic mixtures overyield predominantly when component genotypes differ genetically at a single genomic region on chromosome 2, and that this overyielding is mainly due to a CE.

## An association analysis links the positive diversity effect to the *AtSUC8* gene

The Uk-1 accession was originally collected from the banks of the Dreisam river in the Schwarzwald region of southern Germany. This area is characterized by an edaphic gradient with pH ranging from

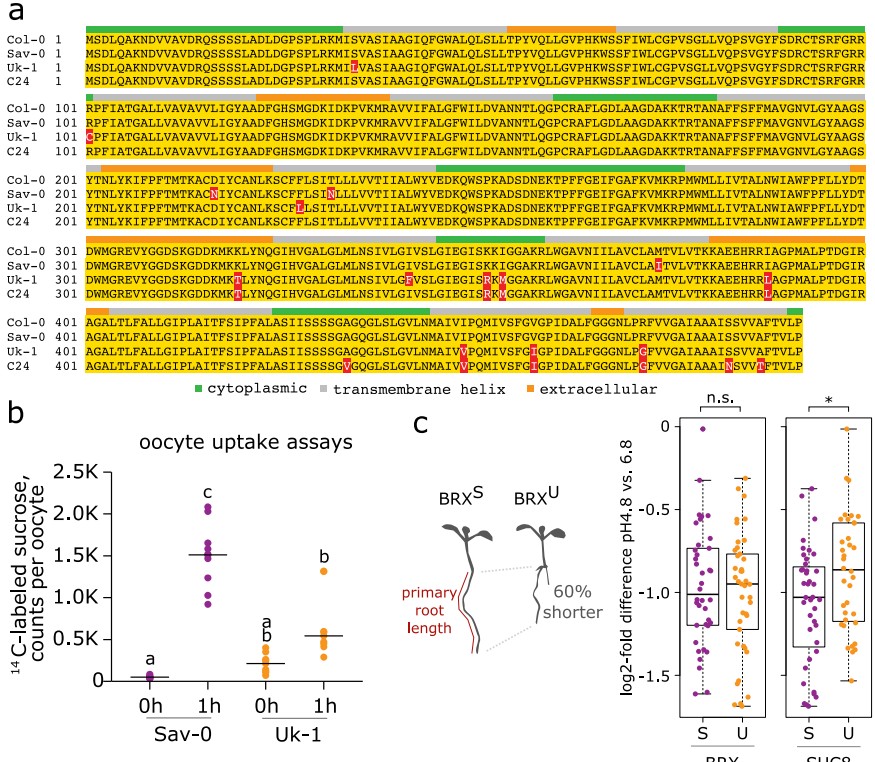

**Fig. 4 | Genetic variation in *AtSUC8* affects protein function and is associated with different root–growth sensitivities to changes in substrate proton concentrations. a** Protein sequence alignments of natural SUC8 variants. Amino acid differences from Col-0 reference sequence are highlighted in red. **b** Sucrose transport activities of the Sav-0 and Uk-1 protein variants in oocytes. Different letters denote significant differences in Tukey's post-hoc contrasts; $n = 9$, except Sav-0, 1 h where $n = 10$. **c** Primary root length differences of genotypes carrying either Sav-0 (S) or Uk-1 (U) alleles at the two loci (*BRX* and *AtSUC8*; number of RILs

($n$): $BRX^S = 38$; $BRX^U = 42$; $AtSUC8^S = 39$; $AtSUC8^U = 33$), grown on agarose plates with different substrate pH. Relative root length of different RILs carrying either allele at the *BRX* (right) or *AtSUC8* locus (left); shown are log2-fold root-length differences of each RIL at pH 4.8 vs. 6.8 (e.g., a log-fold difference of −1 denoting roots being 2-fold shorter at pH 4.8 than at pH 6.8); *$P = < 0.05$ (ANOVA $F_{1,74} = 5.8$; $P = 0.02$); n.s. not significant. Boxes show interquartile ranges with medians; whiskers indicate data ranges up to 1.5 times the interquartile range from the box. Source data are provided as a Source Data file.

neutral to strongly acidic (Supplementary Fig. 3). Previous work has shown that the Uk-1 loss-of-function allele of the *BREVIS RADIX* (*BRX*) gene confers a fitness advantage to plants grown on acidic soil[55] and alters root architecture and plant competition[53,56]. In our experiment, allelic diversity at the *BRX* locus was not, as originally hypothesized, associated with mixture overyielding (Fig. 2b, on lower arm of chromosome 1). Nevertheless, we speculated that the observed overyielding might have been driven by niche complementarity resulting from adaptive divergence along this edaphic gradient. The identified QTL contained 16 protein-coding candidate genes (Supplementary Table S1, putative pseudogenes excluded), including the *Arabidopsis thaliana SUCROSE-PROTON-SYMPORTER 8* (*AtSUC8*), which we considered a candidate diversity-effect gene. The gene encodes a proton symporter that is fueled by the electrochemical gradient across the membrane. *AtSUC8* is predominantly expressed in the root columella[57,58], cells that are in direct contact with the soil, whose pH might affect its activity. To explore the idea that natural genetic variation at the *AtSUC8* locus could drive functional complementarity among *Arabidopsis* genotypes, we re-analyzed previously published data on competition between *Arabidopsis* genotypes[45]. Single individuals of ten tester genotypes (including Sav-0 and Uk-1) each competed separately with each genotype of a panel of 98 natural accessions, in a factorial design (Fig. 3a, b). For each tester-competitor pair, we determined SCAs as in the present study (Methods and Supplementary Fig. 4a, b). We then tested for associations of these SCAs with between-genotype differences at single-nucleotide polymorphisms (SNPs) within the identified QTL on chromosome 2. After adjustment for multiple testing, only one SNP was significantly

associated with a positive diversity effect within the QTL (Fig. 3c, test for differences between mono-allelic and bi-allelic mixture SCAs by linear contrast $t_{947} = 4.1$; $P = 5 \cdot 10^{-5}$, Bonferroni-adjusted $P = 0.007$; standardized effect size = 3.2%). This SNP indeed resides in the *AtSUC8* coding region. Although this is not an unequivocal proof that the identified SNP is the causal genetic polymorphism (it may instead be in tight linkage disequilibrium with the causal one), this finding provides further evidence that genetic differences in or around the *AtSUC8* gene contribute to overyielding of genotype mixtures.

## *AtSUC8* genetic variation affects protein activity and is associated with variation in root plasticity to changes in substrate pH

SUC transporters are highly conserved within and across plant species. Sanger sequencing of the *AtSUC8* alleles from Uk-1, Sav-0, and the reference accession Col-0 confirmed the presence of several non-synonymous SNPs. Compared with the reference allele, the *AtSUC8* coding region of Sav-0 carries three amino acid replacements (one non-conservative), and the Uk-1 allele carries eleven amino acid polymorphisms (six non-conservative) (Fig. 4a). Among the latter, the K320T and the R472G replacements might be functionally relevant, because they also occur in the C24 accession, which we had also used as tester genotype in the association study described above. C24 shares seven amino acid polymorphisms with Uk-1 and shows similar patterns of diversity effects across genotypes (Supplementary Fig. 4c). To determine whether the identified polymorphisms in Uk-1 and Sav-0 affect SUC8 function, we used sucrose uptake assays in a heterologous system to test them functionally. We expressed the

Uk-1 and Sav-0 variants of SUC8 in *Xenopus laevis* oocytes and measured their sucrose uptake kinetics. Whereas SUC8[Sav-0] conferred efficient sucrose uptake as compared to mock-transformed oocytes, significantly lower import was observed with SUC8[Uk-1] (Fig. 4b). We next tested if such functional protein differences also affect root growth under different pH conditions by growing 80 RILs from the Uk-1 × Sav-0 RIL population on two media with pH ~6.8 and ~4.8. For this, we grew seedlings on these media and measured their root length. As expected, root length was reduced (by ≈50%) at low pH and (by ≈60%) in genotypes carrying *BRX*[Uk-1] (Fig. 4c). Relative root length reduction at low pH versus neutral pH did not vary among genotypes carrying different *BRX* alleles (Fig. 4c). However, relative root length reduction was significantly smaller when genotypes carried the *AtSUC8*[Uk-1] instead of the *AtSUC8*[Sav-0] allele (linear model ANOVA $F_{1,74} = 5.8$; $P = 0.02$; Fig. 4c). These findings indicate that Uk-1 carries alleles at multiple loci, including *BRX* and *AtSUC8*, that change root growth and allocation in response to edaphic conditions, in particular environmental proton concentration. Overall, our results thus suggest that genetic differences associated with overyielding of genotype mixtures are related to allele-specific differences in protein and root functioning.

## Discussion

Here, we used two complementary genetic strategies, QTL- and association-mapping, to identify the genetic differences between *Arabidopsis* genotypes that overyielded when grown in mixtures. We found that a large proportion of the overyielding of mixtures of the *Arabidopsis* accessions Sav-0 and Uk-1 was due to allelic diversity at a major-effect QTL on chromosome 2. Two aspects of this QTL mapping study are worth noting. First, our QTL mapping resolution was very high despite using only 18 recombinant lines and their parents. This was due to the competition diallel experimental design, in which genotypes with high-density marker maps were systematically combined into different genotype compositions. Second, although complex traits of individuals, such as growth, are often determined by genetic variants at many loci, each with small effect[59–61], our results - together with findings from recent studies[41,42,44,46] - indicate that complex community-level properties, which depend on interactions between plant individuals, can have simple genetic underpinnings. Thus, our work suggests that positive effects of plant diversity need not be irreducibly complex emergent properties but can have simple causes that are identifiable at the genetic level, even if the mixed genotypes differ at many positions in the genome. Therefore, we suggest that, in order to analyze the causes of overyielding in plant mixtures it is – at least in some cases – easier to first identify the genetic differences responsible for overyielding and then derive the responsible functional traits rather than directly search for the latter using phenotypic differences. In our study system, overyielding seems ultimately driven by allelic variation causing differences in root physiology, a fact we would very unlikely have discovered if we had started with an analysis of phenotypic trait differences used in functional ecology (e.g. specific leaf area, leaf dry matter content, seed size)[19]. The genetic approaches presented here, a further development of earlier approaches presented elsewhere[41,43,44,46,62–64], may thus provide an effective way to understand the propagation of effects across different layers of biological organization, from genes to populations to communities and ecosystems.

Identifying the genes that are important for ecosystem processes may ultimately also be useful to link ecological processes to evolutionary drivers[27,65]. In our study we were able to associate diversity at the *AtSUC8* locus with the overyielding of genotype mixtures. The respective gene encodes a proton-sucrose symporter, i.e., a membrane-associated protein that utilizes a proton gradient to transport sucrose across membranes. The gene is expressed predominantly in root tissues that are in direct contact with the soil.

Genetic differences at the *AtSUC8* locus affect protein function and are also associated with differences in root growth, in a substrate-pH dependent way. Soil chemistry, composition and texture, and the resulting effects on plant–plant interactions are major selective forces, but also important drivers of community structure[66–69]. Consistent with the idea that the Uk-1 genotype exhibits traits that make it better adapted to acidic soil[55], plants carrying the *AtSUC8*[Uk-1] allele showed root growth that was less sensitive so substrate acidification. Although plausible, the hypothesis that evolutionary divergence driven by local adaptation to soil conditions has shaped genetic variation at the *AtSUC8* locus requires further evidence. The identification of this specific gene will facilitate future experiments or population genetic studies on the role of specific nucleotide divergence on root physiology and fitness consequences on different soil types. However, and perhaps surprisingly, genetic variation at the *BRX* locus itself, which had previously been shown to underlie adaptive divergence along this environmental gradient[53,55], did not drive overyielding in our experimental plant stands. Future work should be able to establish possible reasons for these differences between *AtSUC8* and *BRX*, and the specific physiological and morphological effects of the identified genetic variation at the *AtSUC8* locus and their consequences for plant fitness under natural conditions.

A remaining question is how specific aspects of *AtSUC8*-mediated trait differences account for overyielding in genotype mixtures. We think that the different *AtSUC8*-associated responses of root growth to soil acidity promote the partitioning of the physical soil space between plants. One possibility is that pre-existing substrate heterogeneity in soil pH causes niche partitioning among the *AtSUC8* variants, which then results in a more efficient use of the available soil resources[70–72]. Another possibility is that the different pH sensitivity of root growth results in a different root foraging behavior and reduced root competition among genotypes through more complex mechanisms, possibly including soil pH changes that arise from root exudates rather than pre-existing soil heterogeneity. Obviously, there are many different environmental settings or genotype and species combinations for which other traits, related to other genetic differences, may underlie niche partitioning among plants, thus causing overyielding. In each case, the trait-based approaches currently applied to the study of biodiversity effects and overyielding might benefit from gene-based approaches, ultimately not only at the within- but also at the between-species level. On the other hand, our work may offer new ways to design more sustainable cropping systems, in which species or genotype diversity can improve both yield and yield stability in the face of biotic and abiotic stress[48,73–77]. Here, the gene-centered approach may complement currently used trait-centered methods to facilitate the design of high-performance crop variety mixtures.

## Methods

### Germplasm
The Sav-0 and Uk-1 seeds were initially obtained from the Arabidopsis Biological Resource Center at Ohio State University. The Sav-0 × Uk-1 RIL population was described previously[53]. The lines used for the association analysis are described in detail in Wuest et al.[45].

### Plants and growth conditions
Seeds were sown directly on soil and germinated in trays covered with plastic lids under high humidity in a growth chamber at the University of Zurich, Irchel Campus (16 h light, 8 h dark; 20 °C, 60% humidity). The soil substrates are described below. After approximately two weeks, the trays were moved into a greenhouse chamber, where daytime and night-time temperatures were maintained around 20–25 °C and 16–20 °C, respectively. Additional light was provided if required to achieve a photoperiod of 14–16 h. Seedlings were thinned continuously until a single healthy seedling remained per position. The pots were watered *ad libitum*, and, in case of high herbivory pressure

by larvae of the dark-winged fungus gnat, the insecticide ActaraG (Syngenta Agro AG) was applied according to the manufacturer's recommendation. The date of harvesting was determined through the occurrence of 5–10 dehiscent siliques on the earliest flowering genotypes per block. The aboveground plant biomass was dried at 65 °C for at least three days and then weighed.

Assessing accession pair mixtures: nine accession pairs, for which recombinant inbred line populations were publicly available, were chosen for the screen of pair-wise interactions through comparisons of monocultures and two-genotype mixtures. A further pair was chosen based on a large estimate of mixture effects in a previous study[11]. These genotypes of these selected pairs were grown as either monocultures or pair-wise mixtures on different soils and in pots of different sizes as follows: peat-rich soil (Einheitserde ED73, pH -5.8, N 250 mg L$^{-1}$, P$_2$O$_5$ 300 mg L$^{-1}$, 75% organic matter content, Gebrüder Patzer GmbH, Sinntal-Jossa, Germany) and in 6 × 6 × 5.5 cm or 7 × 7 × 8 cm or 9 × 9 × 10 cm pots; sandy soil, a 4:1 mixture of quartz sand:ED73, and in 7 × 7 × 8 cm pots; and Arabidopsis legacy soil, i.e., soil collected from an unrelated previous experiment on which Arabidopsis had grown (originally ED73), and in 7 × 7 × 8 cm pots. Each monoculture or mixture composition in each soil or pot size was grown in each of seven blocks, except those on sand-rich and legacy-soil conditions. The legacy and sandy soil conditions were included only in five of the blocks for logistical reasons. Overyielding of genotypic mixtures containing Sav-0 and Uk-1 was confirmed by growing either (i) four plants in medium sized pots (7 × 7 × 8 cm); (ii) four plants in small pots (5.5 × 5.5 × 6 cm) or (iii) two plants in small pots, all containing ED73 soil. For each pot/density type, 48 mixtures and 24 of each monoculture were sown, treated, and processed as described above. Supplementary Table S2 summarizes these pilot experiments and the corresponding sample sizes across different conditions.

QTL mapping and association study: The QTL-mapping experiment was designed as a half-diallel containing all pair-wise combinations and monocultures of 18 RILs derived from Sav-0 and Uk-1[53] and the two parents (190 genotype mixtures + 20 monocultures). The experiment was conducted in four sequential blocks, with monocultures grown in duplicate within a block, resulting in a total of 920 sown pots over the whole experiment. Pots that did not contain four plants (two per genotype), for example due to seedling establishment problems, were discarded. For the final analysis, data from a total of 808 pots were used. We used soil consisting of a 1:3 mixture of quartz sand:ED73 for the first two blocks. However, because seedling establishment was rather poor on this soil, we changed soil type in blocks three and four to a 3:1 mixture of quartz sand:ED73. Plants were grown and harvested as described above (42–51 days after sowing).

Experimental conditions for the genome-wide association experiment are described in detail elsewhere[45]. In short, the experimental design of the association study consisted of a fully factorial competition treatment of growing ten tester genotypes (Sav-0; Uk-1; Col-0; Sf-2; St-0; C24; Sha; Bay-0; Ler-1; Cvi-0) with each genotype of an association panel of 98 natural Arabidopsis accessions (a subset of the RegMap population[78], including all monocultures) in two replicate blocks. The design thus contained 980 different genotype mixtures and a total of 2154 pots. Each community consisted of two plants (one plant per genotype). Pots were 6 × 6 × 5.5 cm in size and contained a soil mix consisting of four parts ED73 and one part quartz sand. The raw data of the association study are available at https://zenodo.org/record/2659735#.YCt0u2Mo8ml.

## Genotyping and line re-sequencing

For the 18 RIL genotypes used in the QTL-mapping competition diallel analysis, we performed whole-genome resequencing and genotype reconstructions before the genetic analysis. DNA extractions for genome resequencing, library preparation, sequencing and genome reconstruction were done as previously described[42]. In short, reads were aligned to the Arabidopsis reference genome (Col-0 genome, TAIR v10) using BWA version 0.7.16a[79], read sorting and variant calling were performed using SAMtools version 1.5. The genome reconstruction approach broadly followed the method described by Xie and colleagues[80] and is described elsewhere in detail[42]. Raw reads of resequencing the parental accessions Sav-0 and Uk-1 were downloaded from the NCBI SRA homepage (www.ncbi.nlm.nih.gov/sra, SRX011868 and SRX145024). To genotype a wider set of RILs at the AtSUC8 locus (At2g14670 [https://www.arabidopsis.org/servlets/TairObject?id=35253&type=locus]), a Cleaved Amplified Polymorphism (CAPS)-marker assay was developed based on a EcoRV-restriction site in the AtSUC8 coding sequence that is present in the Sav-0 allele but missing in the Uk-1 allele using PCR primers 5′-GGA GAG TGT TGT TAG CCA CGT C-3′and 5′-ACG ATG TGG TAG CTG TAG ATA GAC-3′. DNA extractions for CAPS-genotyping were done using a protocol following Edwards and colleagues[81]. For four RIL genotypes, PCR-genotyping yielded ambiguous results, so we inferred it from flanking markers AtMSQTsnp 123: (Chr 2 pos 1798324) and AtMSQTsnp 138 (Chr 2 pos 8370574)[82]. We also tried to identify RILs that exhibited heterozygosity at the AtSUC8 locus to isolate heterogeneous inbred families, but failed to find any among the 101 RILs screened.

To verify polymorphisms identified in resequencing, Sanger sequencing of the AtSUC8 alleles was performed by amplifying the gene body from genomic DNA using oligonucleotides 5′-ATG AGT GAC CTC CAA GCA AAA AAC GAT-3 and 5′- TTA AGG TAA CAC GGT AAA TGC CAC AAC ACT GC-3′, and further analysed using SnapGene version 6.1. The PCR fragments were then sequenced using those same oligonucleotides as well as oligonucleotide 5′-CAC AAT GAC TAA AGC ATG TGA C-3′. The C24 allele of AtSUC8 was retrieved from published sequence data[83]. Note that because of genomic rearrangements, the gene ID for AtSUC8 (AtC24-2G29550) in the C24 accession differs from that of the other accessions.

## Oocyte uptake assays

Oocyte assays were performed essentially as described[84]. Briefly, the AtSUC8 cDNAs were cloned into pOO2[85]. cRNA was synthesized using the mMESSAGE mMACHINE kit (Lifetechnologies). Oocytes were injected with 50 nL of 150 ng/μL cRNA and incubated in Barth's solution (88 mM NaCl, 1 mM KCl, 2.4 mM NaHCO$_3$, 10 mM HEPES-NaOH, 0.33 mM Ca(NO$_3$)$_2$ × 4 H$_2$O, 0.41 mM CaCl$_2$ × 2 H$_2$O, 0.82 mM MgSO$_4$ × 7 H$_2$O pH7.4) for four days. For uptake experiments, 10 oocytes were kept in 1 ml Barth solution supplemented with $^{14}$C-sucrose at a final concentration of 1 mM or substrate-free control for one hour. Afterwards, oocytes were washed twice in Barth's solution containing gentamycin and were separated into scintillation vials. 100 μl of 10 % SDS (w/v) was added to each scintillation vial and the samples were incubated for 10 minutes. Then, 2 mL of scintillation cocktail (Rotiszint eco plus, Roth, Germany) was added and the vials were vortexed vigorously. Radioactivity was determined by liquid scintillation counting. Experiments were carried out using $^{14}$C-sucrose (536 mCi/mmol, 1 mCi/ml); Hartmann Analytic, Braunschweig, Germany.

## Plate assays and root measurements

Seeds were surface-sterilized with 70% ethanol, followed by 15 minutes in a solution containing 1% bleach and 0.01% Triton-X100 and three sequential H$_2$O washes, then left for stratification at 4 °C overnight. Square MS plates (12 cm) were prepared with 0.8% agarose (instead of agar) and containing 1% sucrose (w/v). The pH was adjusted to 4.5 or 7 using hydrochloric acid or potassium hydroxide and the medium autoclaved. After autoclaving, the measured media pH was again determined (4.8 and 6.8). Six seeds of each of six different genotypes were sown on a plate pair (identical sowing pattern on pH 4.8 and 6.8) and grown in a climate chamber with long-day conditions (16 h light at

20 °C; 8 h dark at 16 °C) for seven days. Plates were scanned twice, once after 4 days and again after 7 days using an EPSON flatbed scanner (model 2450). The primary root length of seedlings was measured using the Fiji software[86].

### Statistical analyses

In the screen for consistently positive pairwise interactions between genotypes, we fitted a linear model of community biomass as function of genotype composition and substrate type (i.e., substrate composition or volume), including a block term. Overyielding of a genotype pair on a given substrate was then estimated as linear contrast between the average monoculture productivity and the mixture productivity (i.e., specifying the contrast matrix K = [−0.5, −0.5, 1], equivalent to the term $1m_{AB} - 0.5m_{AA} - 0.5m_{BB}$ for the case of a monocultures and mixtures of genotypes A and B), using the glht-function of the multcomp-package[87]. A meta-analysis across all Sav-0/Uk-1 mixture experiments was performed using the weighted Fisher method as described previously[88], and using experimental sample sizes as weights (Table S2).

The mapping experiment was performed on two different substrates (two replicated blocks each), and both mean and variance of community productivities differed across substrates. The blocks with more nutrient-rich substrate also had some pots with missing plants due to seedling mortality, which were removed from the analysis. To combine all four blocks for the estimation of SCAs, we first estimated mean community biomass within substrates and calculated SCAs within substrates from average total pot biomass values (BM) as BM = Z*u + SCA whereby Z is the design matrix describing genotype composition of a mixture. To make SCAs comparable across substrates, we divided SCA through the mean pot biomass produced on this substrate. The standardized $SCA_{ij}$ value of a genotype composition (containing genotypes $G_i$ and $G_j$) was then estimated by averaging across substrates. SCA outliers were removed if they differed more than two standard deviations from the population mean in their absolute value. QTL mapping of standardized mixture SCA estimates was then performed by a marker regression approach, where we first fitted a linear model predicting SCAs from allelic composition (3 levels, SS, UU, SU), followed by a contrast between allelic monocultures and mixtures (e.g., $SCA_{SU} - 0.5(SCA_{UU} + SCA_{SS})$), again using the glht function.

A LOD score (−log10(p-value)) of 3 was considered significant, as determined by large-scale simulations[89] under the assumptions: two QTL genotypes ("bi-allelic" and "mono-allelic") and an average chromosome length of 200 cM for Arabidopsis genotype pairs, where recombination events are combined in mixtures. Such a threshold is also in agreement with our previous work comparing this approach to a standard QTL mapping method and a LOD-cutoff based on re-sampling[42]. To estimate overyielding of each mixture and for the additive partitioning, the least square mean productivity estimate was derived for each genotype grown within a composition after adjusting for fixed block effects. A t-test was used to determine if overyielding across all genotype mixtures differed from zero (two-sided t-test with df = 189). The additive partitioning of mixture overyielding into CEs and SEs was done as described in Loreau and Hector[54], and calculated for each genotype mixture separately. Subsequently, a contrast between mixtures that contained two alleles and those that contained one allele only at the QTL on chromosome 2 was tested using the glht-function as described above.

### Analysis of association-study competition experiment

The association study represents a factorial design in which each of ten different genotype (testers) was grown in combination with each of 98 different Arabidopsis genotypes, with all monocultures planted as well. This design was replicated in two blocks. Pots with missing data (e.g., due to seedling mortality) were removed from the analysis. A

genotype's GCA was estimated as described above within each block and values were then averaged across blocks.

Pot biomass depended non-linearly on average genotype GCA (Supplementary Fig. 4). To determine SCAs, we therefore used a quadratic form of the mean GCA to adjust for this non-linearity. Marker regressions on these SCA values for the SNPs within the QTL interval were performed as described above for the QTL mapping approach.

### Reporting summary

Further information on research design is available in the Nature Portfolio Reporting Summary linked to this article.

## Data availability

Data and essential scripts for analyses have been deposited in the Zenodo data repository [https://doi.org/10.5281/zenodo.7896146], and a Source data file is also provided with this paper. The previously published dataset used for the association analysis is also available through the Zenodo data repository [https://doi.org/10.5281/zenodo.6983283]. Sequencing data were deposited in the NCBI Sequence Read Archive, BioProject PRJNA967174. Source data are provided with this paper.

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

## Acknowledgements

We thank Matthias Philipp, Daniela Stöckli and Nicole Ponta for help with plant maintenance and measurements and Matthias Furler for greenhouse support. This work was supported by the University of Zurich, Agroscope, grants from the Swiss National Science Foundation (Ambizione Fellowship PZ00P3_148223) and the University of Zurich Research Priority Program "Evolution in Action" to S.E.W., an Advanced Grant of the European Research Council (AdG #250358) to U.G., and the University of Zurich Research Priority Program "Global Change and Biodiversity" to B.S.

## Author contributions

S.E.W. conducted and analyzed the screen for overyielding of *Arabidopsis* accessions mixtures with support from M.E., and the QTL mapping experiment, with support from B.S. and P.A.N. CSH provided the genotyped Uk-1 × Sav-O RIL population. S.E.W. and N.P. performed the association study with input from U.G.; S.R. and C.S.H. performed the sequence analyses of the *AtSUC8* gene. L.S. and U.H. conducted the oocyte uptake assays. S.E.W. and J.F. performed the root growth experiments. S.E.W., B.S. and U.G. raised funding. S.E.W. together with P.A.N. wrote the first draft of the manuscript. All authors revised and approved the final version of the manuscript.

## Competing interests

The authors declare no competing interests.
