## [Peer Review File · Nature Communications]

Single-gene resolution of diversity-driven overyielding in plant genotype mixturesReviewers' Comments:

Reviewer #1:

Remarks to the Author:

The authors use linkage mapping to identify loci where the presence of different alleles in two neighboring plants results in greater productivity. They then use published association mapping results on neighboring plant interactions to identify a candidate gene in their QTL. They show the two alleles at this locus do differ in their sucrose transport rates in frog oocytes.

Overall I think the study is addressing an important question and is well conducted. I was left unsatisfied with their conclusions as to how this locus would result in overyielding. The authors criticize the trait-based approaches to diversity effects on stand/community level function, but this is precisely a glaring gap in the present study: the functional basis of the diversity effect is not known here.

I also thought there were problems with the framing of the study. I elaborate below:

Selection effects are also important mechanisms of diversity effect on ecosystem function (authors seem to discount this, and overlook it). The authors do not have the ability to quantitatively estimate/distinguish between selection and complementarity effects with their data, so I do not think they should use the term "complementarity" to describe what they found. See Loreau & Hector 2001 Nature.

The individual genotype biomasses in mixture are required to separate these two. Part of the reason that the polyculture yield is greater than the expectation from monoculture may be due to dominance of the more productive genotype in the polyculture ("selection effects").

Also along these lines, the authors do not acknowledge the possibility of negative diversity effects (e.g. Introduction, Figure 1).

The hypothesis of local adaptation underlying the variants here is too unsupported in my view to merit inclusion in the abstract. Additionally, I do not see any justification for the emphasis given to local adaptation, it again comes up in the Discussion:

"However, and perhaps surprisingly, genetic variation at the BRX locus itself, which had previously been shown to underlie adaptive divergence along this environmental gradient" (L344)

I am not familiar with reasons why diversity driven by local adaptation (as opposed to other processes that maintain diversity) should be especially important to overyielding. I have not seen this idea in the literature, and it is not explained here.

In the Introduction's first paragraph, there is a vague and not quite right definition of niche, as "adaptations of species to sets of environmental conditions" (L55)."

"traits as surrogates for niches" (L68) is unclear the meaning here.

The first paragraph of the Introduction ends with an attempt to justify more study of this topic but the authors don't make clear points and argument here.

For example,

"is likely that not a single but many small phenotypic trait differences together determine niche complementarity between plants" (L73)

This would make me pessimistic about the chances of mapping diversity effects to individual loci, though identify individual loci is an aim of this study.

"reductionistic experimental methods" (L79)

This is vague.

"It is reasonable to assume that the mechanisms underlying niche complementarity and overyielding are similar in both cases," (L86)

Is it reasonable? I don't see the authors make this case. If the types of traits that vary among species tend to be different from those that vary within species, then the authors assumption might be incorrect.

I would not refer to groups of plants of the same species communities, as the authors do (e.g. L96, and Figure 1). Communities are by definition groups of species. I would refer to a group of individuals of the same species as a population or in the case of plants as studied here, as a stand.

The authors have previously published on diversity effects on productivity in Arabidopsis, I would like to see the authors explicitly state how this manuscript goes beyond their previous work (Wuest & Niklaus 2018, Wuest et al. 2019).

I do not follow how the authors initially chose AtSUC8 to focus on from among the 16 other protein coding genes in their QTL.

What type of soil was the Wuest et al. 2019 study done in? (the one used in some of the same experiments here?)

The speculated causal mechanism by which AtSUC8 causes overyielding is not clear (paragraph beginning L349). Can the authors clarify? How likely is a microgradient in pH that would somehow restrict a whole plants' root growth?

Additionally, the speculated causal mechanism seems contradicted by the lack of diversity effect at BRX.

Though the authors tested two different soil types (and potentially more it seems from the Methods, e.g. legacy soil, though this was not clear in the Results), they do not seem to use this information to understand their suggested mechanism. It would at least be worth knowing if the effect at AtSUC8 and the broader QTL is soil-type dependent.

Also the importance of pot size is unclear. I hope that analyses accounted for the variable pot sizes used?

"We think that understanding the origins of overyielding may in fact – at least in some cases – be simpler based on genetics than based on traits, where complementarity seems to generally manifest itself as a high-dimensional phenomenon involving a number of different traits (Montazeaud et al., 2020)"

I would argue that genetics only are important via their effects on traits. So I do not see how interactions among traits the result in complementarity (my interpretation of the authors' statement here) would somehow resolve to a simple genetic basis.

Minor:

What does the term "biotope" mean?

L517 - How many SCA outliers were there?

Reviewer #2:

Remarks to the Author:

This is a highly novel paper. By providing an underlying genetic basis, this paper presents a new approach for determining traits responsible for ecological niche differentiation and overyielding within a single plant species. Its implications, though, may be broader. Over 300,000 different plant species coexist on Earth, with hundreds of plant species competing with each other and coexisting in local plant communities. We now know that such coexistence requires evolutionarily unavoidable interspecific tradeoffs. One of the most fundamental tradeoffs occurs at the level of genetic loci. When a locus is occupied by one gene variant, it cannot also be occupied by a different gene variant. By implication, this suggests that the genetic variants that persist in a species would provide fitness advantages for particular types of situations. These polymorphisms and trait differences could thus be the tradeoffs that allow multispecies coexistence and cause ecosystem functioning to depend on both intraspecific and interspecific plant diversity.

This paper's identification of differences in root traits as the likely cause of niche differences and overyielding is also novel. Most ecological studies that seek to understand the implications of interspecific plant trait diversity focus on easily measured leaf morphological or physiological traits. Such analyses, though, have not shed much light on the matter. Even though roots are the way that plants obtain and compete for soil resources such as water, nitrogen, phosphorus and calcium, root traits have generally been ignored because they are difficult to observe and measure. This paper's identification of the importance of root traits, though done indirectly via plant genetics, might spur a much-needed expansion of the plant traits explored for their potential relevance to competition, coexistence, species abundances, and ecosystem functioning.

Here are some issues of concern:

1. The paper's major results are presented in Figures 2 and 3. The legend to Fig 2 C states that "Genotypic mixtures overall exhibit slightly but significantly higher standardized SCAs values than genotypic monocultures." However, no statistical test is provided. Results of a relevant statistical test with P and F values are essential and must be added to the paper. Lines 233-237 do have the relevant stats for Figure 3. These two statistical tests are highly important to demonstrate the validity of the paper's conclusions. This is especially important because lines 120 - 124 state, in seeming contradiction to title of this paper, that there was never any statistically significant overyielding observed in the experiment. This seeming contradiction must be resolved.
2. I am not a plant breeder or molecular geneticist. As best as I can interpret the results as an ecologist, the genetic evidence presented in this paper seems mechanistically deep, highly original and robust, but should be evaluated by an expert.
3. *Arabidopsis thaliana* is a "model" plant species that has been studied in greater depth than any other vascular plant, thus allowing a rigorous exploration of molecular genetics and plant functioning. I do not know how readily this method could be applied to most other plant species.
4. Figure 1 is a conceptual diagram designed to allow readers to more easily understand the paper. However, it took me longer than most readers might be willing to spend to understand it, especially Fig 1 D. In Figure 1 D, I assume that the five sets of parallel lines, colored yellow or blue, are chromosomes. The figure or its legend should state this. Next, Fig 1 D shows recombinant chromosomes, and suggests that some recombinants, when grown in mixture, lead to overyielding, and that this effect is ascribed, in the right-hand side of Fig 1 D to a particular locus, the chromosome region colored either blue or yellow. Readers should not have to guess. Such points should be clarified directly in the legend or on the figure itself.

5. Lines 153 – 182 need some introductory sentences and explanation of terms, especially for terms specific to plant breeding: diallel, specific combining ability and general combining ability. I assume GCA and SCA are based on harvested aboveground biomass of species, but this should be directly stated. Also, this description, and that in the Methods on lines 398 – 404, did not let me visualize how much replication there was of each distinct treatment, how much each soil type was used, and if the four blocks are the same as the four different planting times, or if there were four blocks each time. Please clarify, perhaps using much shorter sentences and a table showing replication for each part of the experimental design.

Response to reviewers, manuscript “Single-gene resolution of diversity-driven community overyielding»

Notes:

Reviewer remarks are in black + italic,

our replies are in blue + normal

Line numbers refer to manuscript version without track changes

Reviewer #1 (Remarks to the Author):

The authors use linkage mapping to identify loci where the presence of different alleles in two neighboring plants results in greater productivity. They then use published association mapping results on neighboring plant interactions to identify a candidate gene in their QTL. They show the two alleles at this locus do differ in their sucrose transport rates in frog oocytes.

1) Overall I think the study is addressing an important question and is well conducted.

We are happy to hear that our work is considered relevant.

I was left unsatisfied with their conclusions as to how this locus would result in overyielding. The authors criticize the trait-based approaches to diversity effects on stand/community level function, but this is precisely a glaring gap in the present study: the functional basis of the diversity effect is not known here.

We agree that it is generally challenging to identify the ultimate mechanistic bases of diversity effects. In our work, we argue that establishing links between different layers of biological organization is an approach that can advance our understanding of such effects. To make this clearer, we have revised Fig. 1b and now illustrate the causal links from genes to proteins to physiology to organisms etc.

The advantage of investigating effects at the genetic level is that the genetic makeup of a plant can be directly manipulated, for example by producing crosses and the recombinant inbred lines we used here. Such manipulations are nearly impossible for most observable functional traits. For example, a typical trait studied in ecophysiology, specific leaf area (SLA), cannot be manipulated directly, and a causal link between SLA differences and diversity effects is therefore very difficult to establish. In contrast, genetic association mapping using appropriate populations comes very close to establishing causality between “genetic traits” (alleles) and the observed diversity effects, although the ultimate proof would of course be a verification by gene editing (which is possible once the specific causal DNA polymorphisms are identified).

Identifying the “genetic cause” already is useful, but we agree that it is desirable to investigate more specifically how these genetic differences propagate to differences in physiology, traits, and plant interactions. In our work, we were able to demonstrate associated differences in enzymatic activities and differences in root growth plasticity to substrate pH. We further find it intriguing that the AtSUC8 protein function depends on a proton gradient, which could hint towards “unusual” niche dimensions underlying niche complementarity. However, it is also clear that we do not yet understand how exactly all this modifies interactions among plants. Therefore, we write (line 368f) “One question that remains open is how specific aspects of SUC8-mediated trait differences account for overyielding in genetically diverse communities.”.

We have now amended the speculation on how complementarity might arise from the identified trait differences as follows (lines 373-384): “We think that the AtSUC8-associated

different responses of root growth to soil acidity somehow promote the partitioning of the physical soil space between plants. One possibility is that pre-existing substrate heterogeneity in soil pH causes niche partitioning among the *AtSUC8* variants, which then results in a more efficient use of the available soil resources⁶²⁻⁶⁴. Another possibility is that the different pH sensitivity of root growth results in a different root foraging behavior and reduced root competition among genotypes through more complex mechanisms, possibly including soil solution pH changes that arise from root exudates rather than pre-existing soil structural heterogeneity. Obviously, there may also be different environmental settings under which other traits, related to other genetic differences, may underlie niche partitioning and complementarity among plants.”

I also thought there were problems with the framing of the study. I elaborate below:

2) Selection effects are also important mechanisms of diversity effect on ecosystem function (authors seem to discount this, and overlook it). The authors do not have the ability to quantitatively estimate/distinguish between selection and complementarity effects with their data, so I do not think they should use the term “complementarity” to describe what they found. See Loreau & Hector 2001 Nature.

The individual genotype biomasses in mixture are required to separate these two. Part of the reason that the polyculture yield is greater than the expectation from monoculture may be due to dominance of the more productive genotype in the polyculture (“selection effects”).

We admit that not including an additive partitioning analysis was a major shortcoming of our initial manuscript, and we thank the reviewer for the suggestion to add it. The reason we had not included the additive partitioning of diversity effects into complementarity (CE) and selection (SE) effects initially was that, in many cases, the genotypes we combined typically had very similar monoculture productivities. This normally precludes large SE because SE are proportional to the covariance of monoculture productivities and deviations of relative yields from expected values. With only little variation in monoculture productivity, this covariance term cannot become large. This starkly contrasts species diversity experiments in which species typically differ much more in size.

However, we agree that it is important to analyze how the different genotypes contributed to the observed diversity effect. In the QTL experiment, we had collected data from individual genotypes in the mixtures, which allowed us to add the Loreau-Hector additive partitioning to the analysis to the revised manuscript. The results show that overyielding (caused by allelic diversity at the QTL) is primarily driven by complementarity effects (CE), and selection effects (SE) were small. Hence, diversity effects were not driven by the dominance of productive genotypes. The analysis and its result is now described in the revised text as follows (Lines 201-213):

“We then applied the additive partitioning method⁴⁶ to test whether overyielding within genotype mixtures (i.e. contrasting bi-allelic with mono-allelic mixtures) was associated with the dominance of productive genotypes carrying either the Sav-0 or Uk-1 allele (a so-called “selection effect”, SE), or whether both allele carriers benefitted from growing in mixture (so-called “complementarity effect”, CE). We found that overyielding was largely driven by CEs (contrast between mono-allelic and bi-allelic mixtures $t_{187} = 2.57$, p-value 0.01) and not by SEs ($t_{187} = 0.201$, p-value 0.84). The average CE was 24.7 mg, which corresponds to 3.1% of the biomass average of all communities in the experiment. Overall, the experiment shows that genotypic mixtures overyield predominantly when component genotypes differ genetically at a single genomic region on chromosome 2, and that this overyielding is mainly due to a statistical complementarity effect.”

Also along these lines, the authors do not acknowledge the possibility of negative diversity effects (e.g. Introduction, Figure 1).

Thank you for pointing this out. Indeed, the possibility of negative diversity effects is often underestimated. However, there are only very few examples that show statistically significantly underyielding of mixtures. Although negative diversity effects were not the focus of our paper, we have included a reference (Montazeaud et al, New Phytologist 2022) that describes negative and positive genetic diversity effects. We also reference an exception to the normally positive genotype diversity - productivity relationships (Bongers et al, New Phytologist, 2020).

The hypothesis of local adaptation underlying the variants here is too unsupported in my view to merit inclusion in the abstract. Additionally, I do not see any justification for the emphasis given to local adaptation, it again comes up in the Discussion: “However, and perhaps surprisingly, genetic variation at the BRX locus itself, which had previously been shown to underlie adaptive divergence along this environmental gradient” (L344). I am not familiar with reasons why diversity driven by local adaptation (as opposed to other processes that maintain diversity) should be especially important to overyielding. I have not seen this idea in the literature, and it is not explained here.

Indeed, this is a key argument we make in our discussion. It rests on two observations that are – individually – undisputed.

First, positive biodiversity effects occur because there is some form of niche complementarity among the plants that are “mixed” in a community. (in line with the paper by Barry et al, 2019, “niche complementarity” and “statistical complementarity effects (CE)” should not be confused: even statistical selection effects (SE) depend on niche differences; with respect to resources, for example, plants must differ in the type or rate of resources use, and hence differ in their niches. The same applies to interactions with other species (e.g. enemies, symbionts) or to environmental tolerances; see Chase and Leibold 2003 for a more detailed discussion, a reference that we now include).

Second, the niches of species (regardless of how a “niche” is specifically defined) reflect the species’ environmental requirements (e.g. climate, edaphic conditions, soil nutrients) and interactions with other organisms (e.g. competitors, consumers, symbionts, pathogens). In our study, we used two ecotypes of *Arabidopsis* that each were locally adapted to their specific environment; specifically, the Umkirch accession is adapted to acidic soils whereas the Slavice accession is adapted to neutral soils – hence, the accessions have diverging niches.

Based on the combination of both lines of logic, we argue that local adaptation is a process that creates niche differences between ecotypes, which in turn can – sometimes, but not necessarily always – lead to overyielding when these divergent ecotypes are grown together in a community. This view is supported by the biochemical function of the AtSUC8 protein (which requires a proton-gradient to transport sucrose, and which differs functionally between protein variants), and by the root growth assays which showed that carriers of the Uk-1 allele at the AtSUC8 locus exhibit less pH-dependent growth suppression than carriers of the Sav-0 allele.

We fully agree that the link between niche complementarity and evolutionary divergence through local adaptation remains a hypothesis that requires further testing. For this reason, we specifically wrote that this remains speculation and that we do not claim generality of this observation (lines 42-44): “We thus speculate that - in the particular case studied here

- evolutionary divergence along an edaphic gradient resulted in the niche complementarity between genotypes that now drives overyielding in mixtures.”.

We also added another disclaimer to the discussion (lines 361-365):

“Although plausible, the hypothesis that evolutionary divergence driven by local soil adaptation to soils has shaped genetic variation at the *AtSUC8* locus requires further evidence. The single-gene resolution will facilitate future experiments or population genetic studies on the role of specific nucleotide divergence on root physiology and fitness consequences on different soil types.”

In the Introduction’s first paragraph, there is a vague and not quite right definition of niche, as “adaptations of species to sets of environmental conditions” (L55).”

We fully agree and have re-worded this statement, which now reads: “Some of these differences represent adaptations of species to sets of environmental conditions ^{5,6}, and the environmental conditions under which a species can maintain a viable population, i.e. its environmental niche space ⁷”

“traits as surrogates for niches” (L68) is unclear the meaning here.

We agree that the whole paragraph on the challenges of trait-based methods in the analysis of biodiversity-community functioning relationships should have been written more clearly, and thank the reviewer for identifying this shortcoming of the initial manuscript.

The changes we made are described in the next answer.

The first paragraph of the Introduction ends with an attempt to justify more study of this topic but the authors don’t make clear points and argument here.

For example, “is likely that not a single but many small phenotypic trait differences together determine niche complementarity between plants” (L73)

This would make me pessimistic about the chances of mapping diversity effects to individual loci, though identify individual loci is an aim of this study.

We fully agree that this is not self-evident. The idea that niche complementarity can only be captured by simultaneously considering many traits, for example using functional diversity indices, is strongly engrained in the biodiversity literature (and also suggested in the paper by Kraft et al, 2015, on species coexistence). However, our study now demonstrates – and quite surprisingly – that a significant fraction of a diversity effect can indeed be mapped to a single locus.

As mentioned above, there are several possible reasons why trait-based methods have struggled so-far to identify causal relationships between phenotypic differences and overyielding in BD-experiments, and multidimensionality is just one of these possibilities. In fact, there has been a recent debate about the utility of traits to explain ecosystem functioning (e.g. reference by van der Plas, NEE 2020, and the now cited “Matters arising” by Hagan et al, 2023 and reply by van der Plas 2023). We have therefore better described some of the challenges of the trait-based methods, and made reference to this ongoing debate. This, we believe, makes it clearer that our paper is relevant in the context of this debate, given that in our case, the BD-effect is caused predominantly by a single QTL and associated with a trait that is likely relatively subtle and difficult to measure (i.e. not multidimensional).

We now re-wrote this whole section as follows (lines 64-77):

[...] Therefore, functional trait differences are often used as indicator of niche differences^{2,5,6,17}, but also this approach has caveats: first, traits often co-vary because of fundamental evolutionary trade-offs between ecological strategies^{18,19}, and this makes it difficult to distinguish correlation from causation in trait-based analyses of biodiversity experiments. Second, while traits often are associated with environmental conditions^{5,20} (e.g. correlation between specific leaf area and soil moisture), it remains unclear whether complementarity in these traits drives overyielding or whether other, unknown, trait differences underlie diversity effects. Finally, it also is possible that not a single but many small phenotypic trait differences together determine niche complementarity among plants^{14,21}. The multivariate nature of phenotypic differences associated with niche complementarity would then make it difficult to identify specific mechanisms that underly biodiversity–productivity relationships^{22,23}. Overall, the link between biodiversity–community functioning relationships and traits is currently not well understood^{4,24,25}. “

“reductionistic experimental methods” (L79)

This is vague.

We have deleted this sentence in the revised text.

“It is reasonable to assume that the mechanisms underlying niche complementarity and overyielding are similar in both cases,” (L86)

Is it reasonable? I don't see the authors make this case. If the types of traits that vary among species tend to be different from those that vary within species, then the authors assumption might be incorrect.

We replaced “mechanisms” with “general mechanisms” and deleted “It is reasonable to assume” and made it clear that here we speculate about the idea that within-species diversity effects are driven by similar mechanisms as between-species diversity effects. We now write (lines 81-84): “The general mechanisms underlying niche complementarity and overyielding may be similar in both cases, although the potential for niche differences between species is clearly greater than between genotypes of the same species.”

I would not refer to groups of plants of the same species communities, as the authors do (e.g. L96, and Figure 1). Communities are by definition groups of species. I would refer to a group of individuals of the same species as a population or in the case of plants as studied here, as a stand.

We agree that our study is special in that the plant communities studied consist of individuals of just a single species. To avoid confusion, we now state this explicitly (lines 85 – 87): “Here, we focus on the study of complementarity among genotypes of the model plant *Arabidopsis thaliana* and on positive genetic diversity effects in model communities consisting of one or two genotypes of this species.”. In addition and for clarity, wherever possible we now use the more specific terms “genotype monoculture”, “genotype mixtures”, or “genotype combinations”.

We kept the term “community” in a few places because it is consistent with its application in the field of community ecology, to which our work is most closely related. In species diversity experiments, monoculture plots (which contain just a single population as the pots in our study) also commonly are referred to as “communities”.

We argue that the term “population” also would have drawbacks. For example, it would imply that there could be further plants species (= additional populations) in the

community, which is not the case. The term population further emphasizes the assemblage of genotypes as larger reproductive unit (interbreeding plants), which is not our focus.

The authors have previously published on diversity effects on productivity in Arabidopsis, I would like to see the authors explicitly state how this manuscript goes beyond their previous work (Wuest & Niklaus 2018, Wuest et al. 2019).

As stated in the manuscript, we published a genetic mapping technique similar to the one we use here in Wuest & Niklaus (2018). In that first paper, we describe the basic concept and methodology for QTL mapping using diallel designs, and show that a diversity-effect QTL can be isolated in a near-isogenic background (“Mendelization”). The work identified a different QTL for a diversity effect that also exhibited different properties. The present study is different and goes far beyond the 2018 publication. It uses a completely different set of genotypes and two independent orthogonal mapping methods. We now resolve the QTL to a single gene, which in a quantitative genetic project normally represents the most challenging step. We further complement this discovery with biochemical and molecular work that targets the physiological functions and morphological changes associated with this allelic diversity effect. We think that this is a major step forward on the way to better analyze and understand diversity effects.

The second study (Wuest et al. 2019, now published in Plos Biology and cited as Wuest et al, 2022) is unrelated to biodiversity effects but analyzes the performance of genotype monocultures instead (as influenced by cooperative traits, not complementarity). In the present manuscript, we re-analyze a dataset that we generated there (with another objective), now with the focus on diversity effects. This analysis allowed us to verify and fine-map the identified diversity effect QTL, using an independent large panel of natural *Arabidopsis* accessions from all over Europe.

I do not follow how the authors initially chose AtSUC8 to focus on from among the 16 other protein coding genes in their QTL.

The initial focus was based on a sophisticated guess, based on functional annotations of the genes within the very well resolved QTL region. First, it was known that Uk-1 exhibited adaptations to acidic soil (BRX-allele). SUC8 is a proton-symporter that is fueled by the electrochemical gradient across the membrane. Expressed at the root-soil interface, it is likely that changes in soil pH will affect the electrochemical gradient, and therefore SUC8 function in vivo, and it is not unlikely that genetic variation in the SUC8 coding region represents adaptations to acidic soil, too. Our initial sophisticated guess was then corroborated by the single-gene resolution of the association analysis (Figure 3).

What type of soil was the Wuest et al. 2019 study done in? (the one used in some of the same experiments here?)

The soil used there was similar in type to the ones used in the current work. We have now included the following information in the methods-section (new lines 458f): “Pots were 6×6×5.5 cm in size and contained a soil mix consisting of four parts ED73 and one part quartz sand.”

The speculated causal mechanism by which AtSUC8 causes overyielding is not clear (paragraph beginning L349). Can the authors clarify? How likely is a microgradient in pH that would somehow restrict a whole plants’ root growth?

What we know is that a large proportion of the observed diversity effect in Uk-1/Sav-0 mixtures is caused by allelic variation in the SUC8 gene. In our opinion this is quite remarkable, because it basically suggests that flipping one or a few nucleotides in the genome is sufficient to “turn the biodiversity effect on or off” (at least a large component of it; see also the comment on trade-offs by reviewer 2). What we also know (from fundamental knowledge of the SUC8 symporter, from the oocyte assays, and from root growth assays in media adjusted to different pH values) is that the function is related to root physiology and likely to proton concentrations around the root. This in our opinion already is very exciting, because it suggests that a very specific root trait is responsible for the observed diversity effect. It is very likely that this trait would never have been considered without the genetic information. Our results thus provide a hint along which “niche dimension” (soil proton concentration) the genotypes are differentiated in some way or another, which ultimately results in mixture overyielding.

How variation in this symporter modifies plant interactions remains unresolved, and we openly admit that we can only speculate about the mechanisms involved. One possibility is that pH microgradients in the soil, or around roots, lead to different root foraging behavior, thereby reducing competition between genotypes. We have amended the discussion section on this point, as mentioned above.

Additionally, the speculated causal mechanism seems contradicted by the lack of diversity effect at BRX.

As we discuss, local adaptation produces ecotypes with diverging niches. However, only a subset of niche differences will support overyielding when these ecotypes are co-cultivated in mixture. Other niche differences will not modify plant interactions in a way that leads to higher community biomass. Similarly, not all genetic differences will promote overyielding. In our study, the differences in BRX gene clearly are adaptive and affect plant interactions and root morphology (as published previously by Gujas et al, Current Biology 2021; and Shindo et al, New Phytologist 2008), but our analyses indicate that diversity in BRX alleles does not promote mixture overyielding. Conversely, diversity in AtSUC8 alleles do promote mixture overyielding.

We believe that this point actually confirms the notion that the trait-based approach to diversity effects is complicated. Prior to our genetic analysis, we expected that differences in BRX gene might be causally related to mixture overyielding (which they weren't).

Though the authors tested two different soil types (and potentially more it seems from the Methods, e.g. legacy soil, though this was not clear in the Results), they do not seem to use this information to understand their suggested mechanism. It would at least be worth knowing if the effect at AtSUC8 and the broader QTL is soil-type dependent.

Also the importance of pot size is unclear. I hope that analyses accounted for the variable pot sizes used?

We initially screened for accession pairs that overyielded when grown in mixture. This screening was carried out using a wide range of soil types, and slightly different planting densities (hence the different pot sizes). However, please note that this initial screening was not meant to be a very rigorous and systematic test of overyielding – the goal was simply to pick a candidate accession pairs that was likely to overyield. The rigorous statistical test for overyielding was carried out later during the systematic investigation of this accession pair, in particular the mapping studies.

Our data (Supplementary Figure 1) show that mixtures of these accessions overyield on all substrates and independently of pot size. We have now added a meta-analysis showing that across all experiments, overyielding was significantly different from zero in Uk-1/Sav-0 mixtures. This analysis is now described in the methods (lines 532 ff: “A meta-analysis across all Sav-0/Uk-1 mixture experiments was performed using the weighted Fisher method as described previously ⁷⁴, and using experimental sample sizes as weights (Table S2).“ and in the results, lines 124 – 127: “Across all experimental settings, mixtures of Sav-0 and Uk-1 yielded an average 5.6% more biomass (range: 0–12%) than expected based on monoculture productivities, and a meta-analysis across all experiments combined revealed that this estimate was significantly different from zero (P = 0.012).“

“We think that understanding the origins of overyielding may in fact – at least in some cases – be simpler based on genetics than based on traits, where complementarity seems to generally manifest itself as a high-dimensional phenomenon involving a number of different traits (Montazeaud et al., 2020)”

I would argue that genetics only are important via their effects on traits. So I do not see how interactions among traits the result in complementarity (my interpretation of the authors’ statement here) would somehow resolve to a simple genetic basis.

We agree that genetic differences affect overyielding only through phenotypic differences. Our study shows that genetic changes leading to differences in root physiology are the ultimate cause of overyielding in the system studied. This discovery is very unlikely to have been made if one had started with the standard “macroscopic” traits often used in functional ecology (e.g. SLA, LDMC, leaf tissue N concentrations, seed size etc). If differences between genotypes can be captured at all by such traits, experience shows that this requires the simultaneous consideration of many traits (i.e. differences between genotypes in higher-dimensional trait space).

To be more clear, we revised this statement as follows (Lines 337-343): “We therefore think that understanding the origins of overyielding may in fact – at least in some cases – be simpler based on genetics than based on the traits usually considered in functional ecology. In our study system, overyielding is ultimately caused by allelic variation causing differences in root physiology, a fact we would very unlikely have discovered starting our analysis from the typically more “macroscopic” traits used in functional ecology (e.g. specific leaf area, leaf dry matter content, seed size) ¹⁹”

Reviewer #2 (Remarks to the Author):

This is a highly novel paper. By providing an underlying genetic basis, this paper presents a new approach for determining traits responsible for ecological niche differentiation and overyielding within a single plant species. Its implications, though, may be broader. Over 300,000 different plant species coexist on Earth, with hundreds of plant species competing with each other and coexisting in local plant communities. We now know that such coexistence requires evolutionarily unavoidable interspecific tradeoffs. One of the most fundamental tradeoffs occurs at the level of genetic loci. When a locus is occupied by one gene variant, it cannot also be occupied by a different gene variant. By implication, this suggests that the genetic variants that persist in a species would provide fitness advantages for particular types of situations. These polymorphisms and trait differences could thus be the tradeoffs that allow multispecies coexistence and cause ecosystem functioning to depend on both intraspecific and interspecific plant diversity.

This paper's identification of differences in root traits as the likely cause of niche differences and overyielding is also novel. Most ecological studies that seek to understand the implications of interspecific plant trait diversity focus on easily measured leaf morphological or physiological traits. Such analyses, though, have not shed much light on the matter. Even though roots are the way that plants obtain and compete for soil resources such as water, nitrogen, phosphorus and calcium, root traits have generally been ignored because they are difficult to observe and measure. This paper's identification of the importance of root traits, though done indirectly via plant genetics, might spur a much-needed expansion of the plant traits explored for their potential relevance to competition, coexistence, species abundances, and ecosystem functioning.

Here are some issues of concern:

*1. The paper's major results are presented in Figures 2 and 3. The legend to Fig 2 C states that "Genotypic mixtures overall exhibit slightly but significantly higher standardized SCAs values than genotypic monocultures." However, no statistical test is provided. Results of a relevant statistical test with *P* and *F* values are essential and must be added to the paper. Lines 233-237 do have the relevant stats for Figure 3. These two statistical tests are highly important to demonstrate the validity of the paper's conclusions. This is especially important because lines 120 – 124 state, in seeming contradiction to title of this paper, that there was never any statistically significant overyielding observed in the experiment. This seeming contradiction must be resolved.*

We agree that the original manuscript was not very clear in this regard.

We have addressed this by, first, providing an additional statistical test showing that the genotype pair screening data already provide evidence for overyielding of the Uk-1/Sav-0 mixture. For this, we performed a meta-analysis across all experiments with this genotype mixture during the "screening" phase. The analysis is now described in the methods (lines 532-534 "A meta-analysis across all Sav-0/Uk-1 mixture experiments was performed using the weighted Fisher method as described previously⁷⁸, and using experimental sample sizes as weights (Table S2)." and in the results section on lines 124 – 127: "Across all experimental settings, mixtures of Sav-0 and Uk-1 yielded an average 5.6% more biomass (range: 0–12%) than expected based on monoculture productivities, and a meta-analysis across all experiments combined revealed that this estimate was significantly different from zero (*P* = 0.012)."

Second, we calculated overyielding in all genotype mixtures of the QTL experiment (i.e. we compare measured mixture productivities to expected values based on monoculture measurements), and found that overall, genotype mixtures exhibit significant overyielding, as stated on lines 171-174: "[...] For each genotype composition, we determined

aboveground dry matter production. Comparing mixture productivities to component monocultures, we estimated that the 190 genotype mixtures overyielded on average by 2.8% (two-sided t-test, $t_{189} = 4.74$, $P < 0.001$).

Note, however, that this comparison of all genotype mixtures to the respective monocultures in the QTL experiment is not a critical comparison (although the difference is significant and provides a “sanity test” for the experiment). The reason is that within the genotype mixtures there are genotype pairs that are effectively “monocultures” with respect to the AtSUC8 locus.

In other words, the relevant test is the comparison of bi-allelic to mono-allelic genotype mixtures (with respect to the AtSUC8 gene), which is highly significant, and stated here (Lines 194-201): “Mixtures that exhibited allelic diversity in this region exhibited a 2.8% (+/- 0.8% s.e.m.) higher SCA than mixtures that contained only one of the two alleles (“mono-allelic” communities, **Figure 2 c**; contrast between mono-allelic and bi-allelic genotype mixtures $t_{187} = 3.53$; $P < 0.001$). The SCA of the mono-allelic genotype mixtures (i.e. mixtures of genotypes which contained only the Sav-0 or only the Uk-1 allele at the identified QTL on chromosome 2) averaged 0.8% higher than the SCA of genotype monocultures, but this difference was not statistically significant.” From this analysis, it can be seen that the QTL on chromosome 2 (around the AtSUC8 locus) explains a major part of overyielding observed across all genotype mixtures.

Third, and in order to extend our understanding of how the chromosome 2 QTL/AtSUC8 locus increases mixture productivity, we now used the Loreau-Hector additive partitioning to show that the complementarity effect is significantly positive (see also comments to reviewer 1, above).

Indeed, a particular strength of our study is that we provide evidence for the same finding from several independent experiments and approaches, whereby the causal genetic differences are resolved to ever finer regions. Specifically, (1) we show that mixtures of the parents overyield, using many different substrates and planting densities in the screen; (2) we QTL-map this overyielding using RIL populations; (3) we GWAS this overyielding in another completely separate type of experiment, using a large panel of Arabidopsis accessions. Hence, we provide independent, orthogonal evidence (using independent genetic material) for the same effect. This “triangulation” provides an additional level of robustness.

2. I am not a plant breeder or molecular geneticist. As best as I can interpret the results as an ecologist, the genetic evidence presented in this paper seems mechanistically deep, highly original and robust, but should be evaluated by an expert.

We indeed think that the interdisciplinary nature of the paper is one of its strengths.

3. Arabidopsis thaliana is a “model” plant species that has been studied in greater depth than any other vascular plant, thus allowing a rigorous exploration of molecular genetics and plant functioning. I do not know how readily this method could be applied to most other plant species.

In principle, the method should be readily applicable for the study of within-species diversity effects, regardless of the species. For example, a similar approach was shown to work in durum wheat in the paper by Montazeaud et al, New Phytologist 2022. Furthermore, the genetic method could be useful for studying diversity effects in species mixture, so long as there is genetic variation in one or both species that affects their interaction.

4. *Figure 1 is a conceptual diagram designed to allow readers to more easily understand the paper. However, it took me longer than most readers might be willing to spend to understand it, especially Fig 1 D. In Figure 1 D, I assume that the five sets of parallel lines, colored yellow or blue, are chromosomes. The figure or its legend should state this. Next, Fig 1 D shows recombinant chromosomes, and suggests that some recombinants, when grown in mixture, lead to overyielding, and that this effect is ascribed, in the right-hand side of Fig 1 D to a particular locus, the chromosome region colored either blue or yellow. Readers should not have to guess. Such points should be clarified directly in the legend or on the figure itself.*

We agree that this figure needed revision and have made two major changes. First, panel b has been revised to better illustrate the differences and commonalities between trait and gene-based approaches, and to show that both must link processes at different level of biological organization. Second, we have made changes to panel 1d, as suggested, to improve readability and clarity.

5. *Lines 153 – 182 need some introductory sentences and explanation of terms, especially for terms specific to plant breeding: diallel, specific combining ability and general combining ability. I assume GCA and SCA are based on harvested aboveground biomass of species, but this should be directly stated. Also, this description, and that in the Methods on lines 398 – 404, did not let me visualize how much replication there was of each distinct treatment, how much each soil type was used, and if the four blocks are the same as the four different planting times, or if there were four blocks each time. Please clarify, perhaps using much shorter sentences and a table showing replication for each part of the experimental design.*

We implemented these suggestions. In the results-section, we now introduce diallel, GCA and SCA as follows:

Lines 154-156: “To study the genetic basis of this effect, we established competition diallels (**Figure 2 a**), an experimental design in which genotypes are systematically combined in all possible pairwise combinations⁴¹⁻⁴⁴.”

Lines 156-162: “In diallels, general and specific combining abilities (GCAs and SCAs, **Figure 2 a**) can be taken as proxies for additive and non-additive mixing properties of genotypes and genotype combinations. GCAs capture the average additive contribution of a genotype to the productivity of mixtures in which it occurs. SCAs capture the productivity deviation of a specific genotype mixture from expectations based on the additive contributions of the components (the sum of the genotypes’ GCAs).”

We also clarified the experiment, overyielding analysis and measurements by now stating (Lines 171-180):

“For each genotype composition, we determined aboveground dry matter production. Comparing mixture productivities to component monocultures, we estimated that the 190 genotype mixtures overyielded on average by 2.8% (two-sided t-test, $t_{189} = 4.74$, $P < 0.001$). This estimate is slightly below the overyielding observed in the parental mixture. However, if the genetic differences driving overyielding are restricted to specific genetic loci, then only part of these genotype mixtures will overyield (the bi-allelic genotype mixtures) while the others will not. Therefore, and in order to genetically map such diversity effect loci, we determined the average SCA across the four blocks for each of the 210 genotype compositions (190 genotype mixtures plus 20 monocultures). [...]”

We have also added a new supplementary table (S2) that highlights the replication levels for the Sav-0/Uk-1 mixture experiments (“screen” and confirmation experiments), which includes overyielding estimates. For the replication in the QTL and association study, where the relevant replication is at the level of the allele (not genotype), we clarified the experimental size more clearly, as follows:

Lines 167-171: “The 20 chosen genotypes were grown in all pair-wise combinations and genotype monocultures were grown in duplicates. The diallel was replicated four times at different dates (temporal blocks), resulting in a total of 920 pots sown. We further used two different substrates (sandy and peaty soils, two blocks each). “

In the methods-section, we added details about the experimental designs as follows:

For the QTL experiment, lines 438-445: “The QTL-mapping experiment was designed as a half-diallel containing all pair-wise combinations and monocultures of 18 RILs derived from Sav-0 and Uk-1⁴⁵ and the two parents (190 genotype mixtures + 20 monocultures). The experiment was conducted in four sequential blocks, with monocultures grown in duplicate within a block, resulting in a total of 920 sown pots over the whole experiment. Pots that did not contain four plants (two per genotype), for example due to seedling establishment problems, were discarded. For the final analysis, data from and a total of 808 pots were used. ”

For the association study, lines 456f: “The design thus contained 980 different genotype mixtures, and a total of 2,154 pots were sown.”

Reviewers' Comments:

Reviewer #1:

Remarks to the Author:

The authors have greatly improved this manuscript.

Overall, the Introduction is more clear and on target than the previous version. I also greatly appreciate the addition of the partitioning of diversity effects.

I have a few important remaining comments about interpretation, however:

The authors do not explain what complementarity is, how it is defined, the general theory behind what could generate it (what kinds of ecological processes), or empirical examples that are known. It is clear that explaining complementarity is important, given the authors state "Here, we focus on the study of complementarity among genotypes" (L85)

My concern is highlighted, for example, by the statement of the authors that refers to general mechanisms but does not explain what they are:

L81 "general mechanisms underlying niche complementarity and overyielding may be similar in both cases although the potential for niche differences between species is clearly greater than between genotypes of the same species"

What exactly is a niche difference? In the context of complementarity, it is not the same as a difference in the classic Hutchinsonian or Grinnellian niche. It is for this reason I am skeptical of the authors' vague claim that alleles locally adapted to different environments should be likely to result in complementarity effects.

I refer the authors to Carroll et al. 2011 Ecology Volume 95 pages: 794-794 and work by Michel Loreau e.g. his monograph "From Populations to Ecosystems" has a couple of chapters on the topic.

This claim in the abstract is not true "Using two orthogonal genetic mapping approaches, we found that between-plant allelic differences at the AtSUC8 locus contribute strongly to mixture overyielding."

The authors used genetic mapping approaches but these cannot conclusively ascribe phenotypic effects to any specific gene. The conclusive difference in the AtSUC8 alleles of interest was demonstrated for sucrose uptake in frog oocytes, not any plant phenotype and definitely not overyielding as claimed in the abstract.

A related problem occurs in the heading "AtSUC8 genetic variation affects protein activity and alter root plasticity to differences in substrate pH" (L275). The protein activity part is correct, but the effect on root pH response is based on RILs that resolve the effect to a QTL containing AtSUC8, but do not resolve the effect to AtSUC8.

Elsewhere in the paper, the authors are appropriately more cautious in their wording.

The definition of an ecological community is universally recognized as multiple species. Using the term here muddies the waters. I recommend the term "stand" as is used in agriculture.

Minor

L170 – Does the 920 pots include the two soil treatments? Or is 920 per treatment?

Reviewer #2:
Remarks to the Author:

I liked the major findings of this paper, and its novelty, during my first review and am pleased that many of my concerns have been addressed in the revised manuscript. My remaining concerns are mainly about the tone of the revised paper. The Abstract and the Introduction still make some unsubstantiated assertions about the "flaws" of niche traits and the "power" of genetic traits. This creates an imaginary conflict for this paper to "solve," but I think that the paper would be stronger if it emphasized that both approaches as important and complementary.

1. For instance, the Abstract states that "In plant communities, diversity often increases productivity and functioning, but the specific underlying drivers are difficult to identify. Most ecological theories attribute the positive diversity effects to complementary niches occupied by different species or genotypes. However, the specific nature of niche complementarity often remains unclear, including how complementarity is expressed in terms of trait differences between plants."

A. T. Clark showed, in an analysis of the Cedar Creek biodiversity experiment, that plant traits of particular mechanistic importance that were directly measured in monocultures correctly predicted multispecies coexistence and the effect of plant diversity on productivity (Clark et al. 2018, Ecology Letters).

2. I agree with the authors that the list of measured plant traits in TRY have often not been very insightful, but this is precisely because they were chosen for ease of measurement and not for their direct link to actual mechanisms of competition. This, though, does not mean that it is impossible to measure actual niche traits. Consider, for instance, the sentence on lines 61-65 which states "An important reason for this knowledge gap is that niche space is almost impossible to quantify directly, and niche complementarity is mostly indirectly implied from observed higher-level phenomena, such as increasing productivity with increasing biodiversity, with little reference to the underlying physiology."

However, numerous experiments, and theory, have shown that traits measured in monocultures, especially the remaining levels of limiting resources (as R^* values), dispersal abilities, and tissue levels of limiting nutrients summarize critical aspects of the nutrient physiology and morphology of a plant species in ways that can mechanistically define niches and that can predict the outcome of competition among several species (for example, Wedin and Tilman 1993; Dybzinski and Tilman 2007; A T Clark et al 2018).

3. Here on lines 72-76, is a clearly correct assertion (the first sentence), that does not necessarily lead to the conclusion in the second sentence "Finally, it also is possible that not a single but many small phenotypic trait differences together determine niche complementarity among plants 14,21. The multivariate nature of phenotypic differences associated with niche complementarity would then make it difficult to identify specific mechanisms that underly biodiversity-productivity relationships 22,23."

The ecology of a plant is the net result of numerous traits. However, that does not mean that it is thus "difficult to identify specific mechanisms that underly biodiversity-productivity relationships." The paper by A T Clark et al. used a few measured monoculture traits of known mechanistic importance, such as the R^* for soil nitrate, that summarize the total effect of numerous other traits on competitive ability for soil nitrate (see Tilman 1990, Chapter 7 in Grace and Tilman, Eds., Perspectives on Plant Competition).

Lines 382-387: I suggest that a bit of caution is needed here, too. All traits have a genetic basis, but

the ecologically relevant traits that underly complementarity may often be determined by multiple genes and interactions among them. Overyielding during intercropping in modern intensive agriculture, for instance, often comes from having two crops that have different climate niches, such as growing best during cool spring weather versus growing best in the heat of the summer. Whole suites of genes are involved in creating such niche differences. The genetic approach used for this paper is an important advance for the root trait it uncovered, but for other traits that are the result of multiple interacting genes, niche-like traits may be more insightful.

The correct plant traits are insightful, just as are the correct genes. Both approaches have great power when properly done. This would be a better overarching theme for the introduction and discussion.

Reviewer #1 (Remarks to the Author):

The authors have greatly improved this manuscript.

Overall, the Introduction is more clear and on target than the previous version. I also greatly appreciate the addition of the partitioning of diversity effects.

I have a few important remaining comments about interpretation, however:

The authors do not explain what complementarity is, how it is defined, the general theory behind what could generate it (what kinds of ecological processes), or empirical examples that are known. It is clear that explaining complementarity is important, given the authors state “Here, we focus on the study of complementarity among genotypes” (L85)

We agree that our previous manuscript was vague about the term complementarity, although it is commonly used to denote differences among plants that underlie the biodiversity–functioning relationship under question. In general, the parts of a system are referred to as “complementary” when they differ in a property that is relevant for the specific aspects of the system that is considered. As also described below, we have substantially re-written sections of the introduction, and avoided the use of “complementarity” overall, in order to avoid confusion (e.g., with the more commonly known term “niche complementarity” or “complementarity effects”).

*For example, we have re-worded the corresponding sentence, which now reads “Here, we focus on positive genetic diversity effects in plant stands of the model plant *Arabidopsis thaliana* and compare genotype mixtures that contain either one or two alleles across regions of the genome.”.*

My concern is highlighted, for example, by the statement of the authors that refers to general mechanisms but does not explain what they are:

L81 “general mechanisms underlying niche complementarity and overyielding may be similar in both cases although the potential for niche differences between species is clearly greater than between genotypes of the same species”

What exactly is a niche difference? In the context of complementarity, it is not the same as a difference in the classic Hutchinsonian or Grinnellian niche. It is for this reason I am skeptical of the authors’ vague claim that alleles locally adapted to different environments should be likely to result in complementarity effects.

I refer the authors to Carroll et al. 2011 Ecology Volume 95 pages: 794-794 and work by Michel Loreau e.g. his monograph “From Populations to Ecosystems” has a couple of chapters on the topic.

This covers several important topics, which we address separately.

1) Niche definition

In our paper, we initially used the niche concept described in the book by Chase & Leibold (“Ecological Niches: Linking Classical and Contemporary Approaches. Biodiversity and Conservation”). This concept integrates the different niche concepts (Elton, Hutchinson, Grinnell, MacArthur) by integrating resource requirements, environmental limitations, and species interactions. However, in our manuscript, we do not aim at further developing theory about niches or niche complementarity. Rather, we want to show that the phenomenon of positive diversity–functioning relationships can be approached empirically through a gene-centered approach. Therefore, our re-written paragraphs of the introduction avoid an extensive treatise of niche theory, and simply state that biodiversity–ecosystem functioning relationships are currently not well understood, and that popular theories attribute them to niche complementarity (in the

references provided). These changes were made in order to avoid side-tracks that are not necessary for the manuscript and the interpretation of its results.

2) Similarity of diversity effects in genotype mixtures and species mixtures

We have edited the relevant sentence to: “The general mechanisms underlying niche complementarity and overyielding may be similar in both cases, although the potential for niche differences between species is greater than between genotypes of the same species.” With this, we simply express that the general mechanisms that promote overyielding (e.g., resource partitioning and other forms of competition reduction) can be expected to occur both in species mixtures and in genotype mixtures. However, because genotypes are more similar than species on average, complementarity is likely also smaller in genotype mixtures than in species mixtures. We think that this is not a controversial statement.

3) Local adaptation leading to trait differences that can underlie complementarity

We presume this comment relates to the statement in the abstract, where we write that “we thus speculate that – in the particular case studied here – evolutionary divergence along an edaphic gradient resulted in the niche complementarity between genotypes that now drives overyielding in mixtures.”

Here, we speculate that in the specific case we studied, adaptation of *Arabidopsis* to different soil acidities caused the difference in root traits that – when genotypes are grown in mixtures – results in overyielding. This speculation rests on (1) the fact that the Uk-1 accession was collected from a region with acidic soils (Figure S3) and contains alleles that confer fitness advantages relative to Sav-0 when grown on acidic soil (literature cited); (2) our experiments showed that the Uk-1 allele found at the locus associated with the observed overyielding effect was related to a pH-dependent root growth response, is expressed in roots, and that the corresponding protein is functionally dependent on a proton-gradient across the membrane. Note that we do not suggest that local adaptation to different conditions automatically causes complementarity that results in overyielding (in fact, the case of *BREVIS RADIX*, which we discuss in the paper, is an example to the contrary), but we think that this is an interesting possibility and that, in this particular case (*AtSUC8*), there is good evidence for it. Also, we explicitly declare that this is speculative, and may only apply to this specific case. All our points are based on empirical findings, not on theoretical considerations.

In summary, we use niche complementarity as it is used in the references we cite and independent of the way it came about, i.e., via evolutionary niche differentiation within a population or via niche separation by diversifying evolution among populations.

This claim in the abstract is not true “Using two orthogonal genetic mapping approaches, we found that between-plant allelic differences at the *AtSUC8* locus contribute strongly to mixture overyielding.”

The authors used genetic mapping approaches but these cannot conclusively ascribe phenotypic effects to any specific gene. The conclusive difference in the *AtSUC8* alleles of interest was demonstrated for sucrose uptake in frog oocytes, not any plant phenotype and definitely not overyielding as claimed in the abstract.

We agree with Reviewer #1 and have changed the respective sentence as follows: “Using two orthogonal genetic mapping approaches, we find that between-plant allelic differences at the *AtSUC8* locus are strongly associated with mixture overyielding.”

A related problem occurs in the heading “*AtSUC8* genetic variation affects protein activity and alter root plasticity to differences in substrate pH” (L275). The protein activity part is correct, but the

effect on root pH response is based on RILs that resolve the effect to a QTL containing AtSUC8, but do not resolve the effect to AtSUC8. Elsewhere in the paper, the authors are appropriately more cautious in their wording.

The heading has now been changed to: “AtSUC8 genetic variation affects protein activity and is associated with variation in root plasticity to changes in substrate pH”

The definition of an ecological community is universally recognized as multiple species. Using the term here muddies the waters. I recommend the term “stand” as is used in agriculture.

We have now replaced “communities” with “stands”, “genotype mixtures” or “genotype compositions” throughout the text when referring to our experiment. In the revised manuscript, we only refer to communities in the introduction and in reference to the broader ecological context and previous work with different species.

Minor

L170 – Does the 920 pots include the two soil treatments? Or is 920 per treatment?

We have now clarified in the main text that the 920 pots refer to the whole experiment.

Reviewer #2 (Remarks to the Author):

I liked the major findings of this paper, and its novelty, during my first review and am pleased that many of my concerns have been addressed in the revised manuscript. My remaining concerns are mainly about the tone of the revised paper. The Abstract and the Introduction still make some unsubstantiated assertions about the “flaws” of niche traits and the “power” of genetic traits. This creates an imaginary conflict for this paper to “solve,” but I think that the paper would be stronger if it emphasized that both approaches as important and complementary.

We thank Reviewer #2 for pointing this out. We now realize that our word choice was, in some cases, unfortunate in this respect and have therefore re-written substantial parts of the introduction (see also responses above). By no means do we want to suggest that trait-based approaches are flawed. We have already tried to avoid this in our previous version of the manuscript, but now make it more explicit by stating: “In functional ecology, trait differences are often used as an indicator of niche differences, and are major determinants of the composition, diversity, and functioning of communities.”

We argue that trait-based approaches are difficult to apply for the study of biodiversity effects, although we now include some successful examples, such as those indicated below. We still write that the drivers of positive biodiversity–ecosystem functioning relationships are currently not well understood and think that there is likely not a single approach that will solve this question. Gene-centered approaches can provide additional information to trait-based methods. Reflecting this, we now end our conclusion with the statement: “Here, the gene-centered approach may complement currently used trait-centered methods to facilitate the design of high-performance crop variety mixtures.”

In the revised manuscript, we have embraced the reviewer’s suggestion to make this point more explicit (see also last point, below). We have now emphasized this point in the re-written paragraphs of the introduction, for example, as follows: “Here, we explore a gene-based approach to investigate the causes of positive diversity–productivity relationships in plant stands. Our new approach complements traditional trait-based methods and helps to identify causal drivers.”

1. For instance, the Abstract states that “In plant communities, diversity often increases productivity and functioning, but the specific underlying drivers are difficult to identify. Most ecological theories attribute the positive diversity effects to complementary niches occupied by different species or genotypes. However, the specific nature of niche complementarity often remains unclear, including how complementarity is expressed in terms of trait differences between plants.”

A. T. Clark showed, in an analysis of the Cedar Creek biodiversity experiment, that plant traits of particular mechanistic importance that were directly measured in monocultures correctly predicted multispecies coexistence and the effect of plant diversity on productivity (Clark et al. 2018, Ecology Letters).

We thank Reviewer #2 for this useful reference, which we have now included in the manuscript. As stated above, we also have re-written the introduction to better reflect the current state of knowledge and to avoid introducing an unnecessary controversy around the merits of trait-based ecology. We have re-worded the sentence as follows: “However, it is currently less clear which trait differences drive the positive BEF relationships in plant communities^{17,25–27, but see 22,28–31}.”

2. I agree with the authors that the list of measured plant traits in TRY have often not been very insightful, but this is precisely because they were chosen for ease of measurement and not for their direct link to actual mechanisms of competition. This, though, does not mean that it is impossible to measure actual niche traits. Consider, for instance, the sentence on lines 61-65 which states “An important reason for this knowledge gap is that niche space is almost impossible to quantify directly, and niche complementarity is mostly indirectly implied from observed higher-level phenomena, such as increasing productivity with increasing biodiversity, with little reference to the underlying physiology.”

However, numerous experiments, and theory, have shown that traits measured in monocultures, especially the remaining levels of limiting resources (as R^* values), dispersal abilities, and tissue levels of limiting nutrients summarize critical aspects of the nutrient physiology and morphology of a plant species in ways that can mechanistically define niches and that can predict the outcome of competition among several species (for example, Wedin and Tilman 1993; Dybzinski and Tilman 2007; A T Clark et al 2018).

Thank you for this clarification. We have toned this down and added references to the successful use of traits as shown in the previous response.

3. Here on lines 72-76, is a clearly correct assertion (the first sentence), that does not necessarily lead to the conclusion in the second sentence “Finally, it also is possible that not a single but many small phenotypic trait differences together determine niche complementarity among plants 14,21. The multivariate nature of phenotypic differences associated with niche complementarity would then make it difficult to identify specific mechanisms that underly biodiversity–productivity relationships 22,23.”

This sentence was indeed misleading. We have now revised it to read: “Finally, it also is possible that many small phenotypic trait differences need to be considered simultaneously to adequately capture niche complementarity between plants, making it more difficult to identify specific mechanisms that cause BEF relationships”

The ecology of a plant is the net result of numerous traits. However, that does not mean that it is thus “difficult to identify specific mechanisms that underlie biodiversity–productivity relationships.” The paper by A T Clark et al. used a few measured monoculture traits of known mechanistic importance, such as the R^* for soil nitrate, that summarize the total effect of numerous other traits on competitive ability for soil nitrate (see Tilman 1990, Chapter 7 in Grace and Tilman, Eds., Perspectives on Plant Competition).

This reference indeed is a great example for the use of traits in ecology. Our point merely is that traits often are highly correlated, and that – because the decisive trait is often not measured – effects are approximated with larger sets of traits that correlate with the unknown trait, by chance or for other reasons. This makes it difficult to assign cause–effect relationships, unless trait differences could be “randomized” across species compositions at a given diversity level. This is often impossible, due to the species’ evolutionary history, which cannot be “undone” or “randomized”. At the same time, this is clearly a merit of the genetic methods, because genes can indeed be “randomized” across new genotypes by crossing/sexual recombination, and then re-assembling these genotypes into new mixtures.

Lines 382-387: I suggest that a bit of caution is needed here, too. All traits have a genetic basis, but the ecologically relevant traits that underly complementarity may often be determined by multiple genes and interactions among them. Overyielding during intercropping in modern intensive agriculture, for instance, often comes from having two crops that have different climate niches, such as growing best during cool spring weather versus growing best in the heat of the summer. Whole suites of genes are involved in creating such niche differences. The genetic approach used for this paper is an important advance for the root trait it uncovered, but for other traits that are the result of multiple interacting genes, niche-like traits may be more insightful. The correct plant traits are insightful, just as are the correct genes. Both approaches have great power when properly done. This would be a better overarching theme for the introduction and discussion.

Traits of course have a genetic basis, and we emphasize this fact in Figure 1b. The genetic basis of biodiversity effects, however, are currently essentially unknown. In the cases in which we have done this (for example, in the present work, or in Wuest & Niklaus 2018), we – perhaps surprisingly – have found that the observed complementarity effects had an extremely simple genetic architecture. Prior to performing these investigations, we tended to assume that it would have a more polygenic basis, but, in the meantime, we have become more cautious with such statements. As we state in the introduction and conclusion, trait-based approaches have their merits but there also are difficulties in applying them to the study of biodiversity effects. Genetic approaches – the topic of the present paper – may support more traditional methods and can provide novel insights into biodiversity effects. We think that the two approaches ultimately should be combined: genetic approaches can help identifying novel traits (such as the root-related traits in this manuscript), and studying these (e.g., root physiology, root growth, foraging and competition, and consequences from plants to communities) using functional ecology approaches should then allow for a more functional understanding. Importantly, the genetic approach is, of course, also applicable if multiple genetic loci are associated with mixture overyielding as in the case you mention in your comment, and as indeed we had expected at the start of our work.

As shown above, we have re-written the introduction to support this notion that there is likely not a single method that is superior, but each one has its advantages and limitations.